# The 2010-2015 mega drought in Central Chile:
# Impacts on regional hydroclimate and vegetation

René Garreaud[1,2,*], Camila Alvarez-Garreton[3,2], Jonathan Barichivich[3,2], Juan Pablo Boisier[1,2], Duncan Christie[3,2], Mauricio Galleguillos[4,2], Carlos LeQuesne[3], James McPhee[5], Mauricio Zambrano-Bigiarini[6,2]

[1]Department of Geophysics, Universidad de Chile, Santiago-Chile
[2]Center for Climate and Resilience Research (CR2), Santiago-Chile
[3]Laboratorio de Dendrocronología y Cambio Global, Instituto de Conservación Biodiversidad y Territorio, Universidad Austral de Chile, Valdivia-Chile.
[4]Faculty of Agronomic Sciences, Universidad de Chile, Santiago-Chile
[5]Department of Civil Engineering, Universidad de Chile, Santiago-Chile
[6]Department of Civil Engineering, Faculty of Engineering and Sciences, Universidad de La Frontera, Temuco-Chile

*Correspondence to*: René. D. Garreaud (rgarreau@dgf.uchile.cl)

**Abstract.** Since 2010 an uninterrupted sequence of dry years, with annual rainfall deficits ranging from 25 to 45%, has prevailed in Central Chile (western South America, 30-38°S). Although intense 1- or 2-year droughts are recurrent in this Mediterranean-like region, the ongoing event stands out because of its longevity and large extent. The extraordinary character of the so-called Central Chile mega drought (MD) was established against century long historical records and a millennial tree-ring reconstruction of regional precipitation. The largest MD-averaged rainfall relative anomalies occurred in the northern, semi-arid sector of central Chile but the event was unprecedented to the south of 35°S. ENSO neutral conditions have prevailed since 2011 (but for the strong El Niño 2015) contrasting with La Niña conditions that often accompanied past drougths. The precipitation deficit diminished the Andean snowpack and resulted in amplified declines (up to 90%) of river flow, reservoir volumes and groundwater levels along central Chile and westernmost Argentina. In some semiarid basins we found a decrease in the runoff-to-rainfall coefficient. A substantial decrease in vegetation productivity occurred in the shrubland-dominated, northern sector, but a mix of greening and browning patches occurred farther south where irrigated croplands and exotic forest plantations dominate. The ongoing warming in central Chile, making the MD one of the warmest 6-year period on record, may have also contributed to such complex vegetation changes by increasing potential evapotranspiration. We also report some of the measures taken by the central government to relieve the MD effects and the public perception of this event. The understanding of the nature and biophysical impacts of the MD helps as a foundation for preparedness efforts to face a dry, warm future regional climate scenario.

## 1 Introduction

Droughts have been recognized as a major climate hazard in many regions worldwide (e.g., Mishra and Singh, 2010; Seneviratne et al., 2012). Depending on its duration and intensity, a lower-than-average precipitation condition (i.e., a meteorological drought) can lead to a substantial decrease in surface water resources, soil moisture and groundwater, thus causing a multiplicity of adverse ecological, social and economic impacts (see a review in Wilhite, 2000). Semi-arid, Mediterranean-like regions are particularly prone to droughts given that most of the annual rainfall accumulation is accounted in a few events so that individual missed storms can have significant impact (e.g., Ragab and Prudhomme, 2002; Rockstrom et al., 2010).

Over the last decades, subtropical land-areas have experienced droughts that are not only intense (as per the annual rainfall deficit) but also protracted, greatly increasing the accumulated magnitude and impacts of these events (Dai, 2011, 2013; Schubert et al., 2016). There is a vast body of literature describing the nature of a recent multi-year (2012-2014) drought in California (e.g., Swain, 2015; Williams et al., 2014; Griffin and Anchukaitis ,2014) and its unprecedented effects on hydrology, forest fires and agriculture (e.g., AghaKouchak et al., 2014; Mao et al., 2015). A severe, protracted drought also afflicted southeastern Australia in the recent past (ca. 1997-2009, Cai et al., 2014; Saft et al., 2016). The winter rainfall deficit during the so-called Millennium Drought caused major water crises, affecting river ecosystems and agricultural production as reviewed in van Dijk et al., (2013). Intense rainfall deficits have also prevailed recently in land areas surrounding the Mediterranean Sea (Garcia-Herrera et al., 2007; Hoerling et al., 2012), China (Barriopedro et al., 2012), South Africa (Rouault and Richard, 2003) and the Middle East (Trigo et al., 2010; Kelly et al., 2005).

Central Chile, the narrow strip of land between the southeast Pacific Ocean and the Andes cordillera (30°-38°S), features an archetypical Mediterranean climate (e.g. Millerh, 1976; see also section 3) with annual mean precipitation ranging between 100 and 1000 mm and a marked seasonal cycle. Rainfall exhibits substantial interannual variability historically associated with El Niño Southern Oscillation (ENSO, e.g., Aceituno 1988; Montecinos et al., 2000; Montecinos and Aceituno 2003). Since the early 1980s a precipitation decline is evident along the coast (Quintana and Aceituno, 2012) and the Andes cordillera (Masiokas et al., 2016), accentuated by an uninterrupted rainfall deficit since 2010 to date. Boisier et al., (2016) made use of historical Global Circulation Model results to establish that SST-forced circulation changes account for only about half of the precipitation trend.

The recent (2010-2015) multi-year, regional-scale dry event has been referred to as the Central Chile megadrought (MD, a term coined by CR2, 2015). A preliminary survey conducted by our group found significant impacts of this protracted drought on surface hydrology, groundwater, sediment exportation into the ocean, vegetation and fire activity along Central Chile (CR2, 2015), and we show later that such multi-year drought is already unprecedented in the historical record and quite unusual in the last millennium. Between 2010 and 2015 the Chilean national authorities decreed emergency conditions

in seven (out of 15) administrative regions, applying exceptional water-management measures and lending relief packaged to local communities (CR2, 2015). By depleting the subtropical Andes snowpack, this drought also reduced water resources in western Argentina (Bianchi et al., 2016; Rivera et al., 2017).

The goal of this work is to place the Central Chile MD in a historical and long-term context, describe the concomitant large-scale circulation anomalies and document its main impacts on hydrology and vegetation productivity. These issues have been only partially addressed in technical reports (CR2, 2015; MOP, 2015) and a recent paper that focuses on the long term drying trend along the west coast of South America (Boisier et al., 2016), of which the mega drought is the latest manifestation. On the other hand, the overall goal of the present contribution is a particularly relevant task given the prospects of climate change in this region for the 21st century. Model-based climate projections consistently indicate a reduction in mean annual precipitation (up to 30% relative to current values) and an increase in surface air temperature (up to 4°C at the top of the Andes) for the 2070–2100 period under high emission scenarios (A2 in Fuenzalida et al., 2007, RCP8.5 in Bozkurt et al., 2017), severely disrupting agriculture, hydropower generation and availability of drinking water (Vicuña et al., 2010) in this already water-stressed area (MOP 2013). In this context, the protracted and spatially extensive precipitation deficit presently occurring offers an analog of the region's future from which key lessons may be learned. On a broader perspective, the MD seems to differ from the intense but short-lived droughts that characterize Central Chile's climate and may lead to environmental effects that have not been observed before. Describing those effects will shed light on the functioning of the atmosphere-hydrosphere-biosphere system in a Mediterranean-like region under extreme events.

The paper structure is as follows. In section 2 we describe station-based datasets of rainfall, river flow and temperature, a millennial tree-ring based reconstruction of precipitation for Central Chile, and satellite derived products of snow cover and water equivalent, potential evapotranspiration and vegetation productivity. A brief description of the climate conditions in Central Chile is provided in section 3. A regional precipitation index and the standardized precipitation index in hundreds of stations along the region are employed in section 4 to identify previous droughts helping to place the MD in context. The rainfall deficit during the MD is described in section 5, where we also asses its recurrence. In that section we briefly describe the large-scale conditions accompanying the MD whose underlying causes are still under study; the long-term drying trend upon which the MD is superimposed has been described in Boisier et al., (2016). In section 6 we describe the MD impacts in hydrology (streamflow, snow accumulation, potential evapotranspiration) and vegetation (plant productivity). In the discussion (section 7) we describe some of the measures taken by the central government to provide relief for the MD effects across central Chile and their public perception, and identify future research lines. Our main findings are summarized in section 8.

## 2 Datasets

### 2.1 Station records and gridded precipitation

The Chilean directorate of water resources (DGA) and the National Weather Service (DMC) maintain more than 700 rain gauges along Chile. Almost all the stations are conventional pluviometers, and began their measurements in the 60's operating until now. Of particular relevance, seven stations have continuous data from 1915 onwards (Table 1). From the original daily observations we computed monthly accumulations if less than 5% of the days were missing, thus retaining

nearly 300 stations (Figure 1) between 30°-38°S. Likewise, we use near-complete monthly records of extreme temperature in 102 stations and (mostly unimpaired) river flow in 119 stations. Station data are available from the Chilean Climate Explorer (http://explorador.cr2.cl). The reference period we used is 1980-2010 unless otherwise noted.

For the calculation of the standardized precipitation index (section 4) we use a gap-filled version (Boisier et al., 2016) of the original station data that include continuous monthly records from 1960-2015 in 153 stations between 32.5°S and 36.5°S (the

core of central Chile). For the analysis of the drought impact on hydrology and vegetation we use a gridded precipitation dataset, recently developed within the CR2 activities. This product, available from 1979 onwards on a ~5×5 km$^2$ (0.05° lat-lon) grid, results from a merge of a statistical downscaling of large-scale data provided by the ERA-Interim reanalysis data (Dee et al., 2011) and an interpolation of the quality-controlled station-based rainfall records (for further information on this product please visit http://www.cr2.cl/recursos-y-publicaciones/bases-de-datos/).

**2.2 Tree-ring based rainfall reconstruction**

Tree-rings from the long-lived coniferous tree *Austrocedrus chilensis* represent the best annually resolved proxy for reconstructing multicentury precipitation variability in subtropical South America (Le Quesne et al., 2006; 2009; Christie et al., 2011). Here we develop a new tree-ring based precipitation reconstruction for Central Chile utilizing the existing *A. chilensis* samples of El Asiento (ELA, 32.4°S-70.5°W) and Agua de la Muerte-El Baule (ELBAMU, 34.3°S-70.3°W) sites

(LeQuesne et al., 2006; 2009) and updating collections from living and sub fossil wood, resulting in a massively replicated record from 502 tree-ring series encompassing the entire last millennium. The ring width measurements of the two sites were detrended, prewhitened and site chronologies were calculated as the robust biweight mean of the tree-ring indices, finally producing a regional tree-ring record from the average of the two site chronologies. The expressed population signals (EPS) of each site chronology were all above 0.85 threshold (i.e., 85% common signal and 15% noise) across its length. The

number of tree-ring series at year 1000 AD and for the entire chronology were 34 and 180 for ELA and 29 and 322 for ELBAMU, respectively.

Our precipitation reconstruction was developed by calibrating the regional tree-ring chronology with the updated regional precipitation record (1930-2014) of Central Chile used by LeQuesne et al., (2006), utilizing standard methods in dendroclimatology (Cook and Kairiukštis, 1990). Our annual precipitation target was the June-December (winter-spring)

period when more than half of the total annual precipitation occurs (Fig. 1b). As there is no significant low-order autocorrelation in the precipitation target, white noise versions of the tree-ring chronologies were used to develop the regional tree-ring chronology. A bivariate linear regression was used to calibrate the regional tree-ring chronology on the logarithm values of the instrumental target for the period 1930–2014. The reconstruction model was developed using the

leave-one-out cross-validation procedure. The regression model explains 53% of the rainfall variance (F = 93.4; $p < 0.001$) and verifies satisfactorily (Reduction of Error statistic 0.5). The correlation between the back-transformed reconstructed and observed precipitation was 0.67 ($n = 85$; $p < 0.001$).

## 2.3 Satellite derived products

We employed a snow water equivalent (SWE) distributed reconstruction developed by Cornwell et al., (2016) over the subtropical Andes for the period 2001-2015, fully independent from precipitation data. The SWE reconstruction combines information on fractional snow cover depletion obtained from the Moderate Resolution Imaging Spectroradiometer (MODIS) instrument on board the Terra and Aqua satellites (MOD10A1, Hall et al., 2010) and a physically based energy balance model that explicitly computes shortwave and long wave radiation, and adds a simplified parameterization of

turbulent fluxes. The resulting product is a daily estimation of SWE over a $500 \times 500$ m$^2$ grid from August 15th to the last day of the next year in which snow cover is present at each pixel.

Potential Evapotranspiration (PET) is assessed through the MODIS product MOD16 (Mu et al., 2011) available monthly from January 2000 to December 2014 on a $1\times1$ km$^2$ grid. PET represents the energy available to evaporate water given no surface water limitations (de Jong et al., 2013). PET minus rainfall difference gives a realistic estimate of the water deficit

due to climate (e.g., Tsakiris and Vangelis, 2005) and several studies have shown evidence that augmented PET does have a detrimental effect on vegetation (Vicente-Serrano et al., 2013). The algorithm used in MOD16 is based on the Penman-Monteith approach and detailed information about the formulation can be found in Mu et al., (2011).

The response of vegetation productivity to the MD was evaluated using 16-day Terra MODIS Enhanced Vegetation Index (EVI) from Collection 6 MOD13C1 product ($5\times5$ km$^2$) during the period 2001–2015. EVI is a proxy of canopy

photosynthetic capacity (Huete et al., 2002) and several studies have found a strong correlation between EVI and gross primary productivity in a wide range of vegetation types (e.g., Sims et al., 2006). For further analysis of changes in vegetation, we use a national Land Cover product (Zhao et al., 2016) which is a based on the FAO land cover classes (Di Gregorio 2005). The product describes the land cover categories in 2014 and has a 30-m spatial resolution.

## 3 Study region and climate context

Central Chile (30-38°S), in western South America, features a narrow (~200 km wide) strip of lowlands bounded to the east by the Andes cordillera that reaches more than 4000 m ASL in this range of latitudes (Fig. 1). This region hosts over 9 million inhabitants (nearly two thirds of the Chilean population), major cities (including Santiago, the Chilean capital) and key economic activities (e.g., mining, agriculture, timber production and hydropower generation). To the north of Santiago (33°S) water demands in recent years are equal to or larger than water availability (Hearne and Donoso, 2005). This semiarid

portion of Central Chile is referred to as the northern region; the portion to the south of 34°S is referred to as the southern region.

The Mediterranean-like climate of Central Chile (e.g. Miller, 1976) is dictated by the subtropical anticyclone and the storm track at midlatitudes, resulting in a marked meridional gradient in annual mean accumulation (Fig. 1a). Near the coast around 30°S annual mean rainfall is below 100 mm and concentrated almost exclusively during the austral winter (JJA, Fig. 1b). In contrast, annual mean values above 1000 mm are observed to the south of 37°S where rainfall events are still more frequent in the winter season but can also occur during summer (Figs. 1a, 1b). Orographic precipitation enhancement over the windward slope of the Andes cordillera further creates a west-east increase in precipitation. The zonal gradient is barely visible in Fig. 1a, because the color scale favors the visualization of the meridional gradient and the lack of long-term records high in the Andes, but can increase the precipitation by a factor 2-3 between the lowlands and the windward slopes (Viale and Garreaud 2015). In addition to the gradients in annual mean precipitation, the coefficient of variation changes markedly along Central Chile, from ~100% around 30°S, ~50% at 33°S and ≤25% to the south of 37°S (Fig. 1c).

The influence of El Niño-Southern Oscillation (ENSO) on climate variability in Central Chile is strong and involves a warm–wet/cold–dry relation (e.g., Aceituno, 1988; Montecinos et al., 2000; Montecinos and Aceituno, 2003). During La Niña years, a poleward shift of the storm track (Solman and Menendez, 2002) along with higher pressures in the subtropical Pacific and weaker mid-level westerlies conspire to produce a rainfall deficit in Central Chile. Roughly the opposite large-scale configuration takes place during El Niño years often associated with above normal winter precipitation due to frequent high-latitude blocking and widespread weakening of the subtropical anticyclone (Montecinos and Aceituno 2003).

With the exception of a relatively small glaciated area the Andes of central Chile are virtually free of snow by late summer/early fall. Successive storms during winter (average freezing level of 2300 m ASL; Garreaud 2013) build-up a seasonal snow pack that reaches its maximum extent in late winter (August-September) as illustrated for the upper Maipo river basin (east of Santiago) in Fig. 2a. For reference, the September 1st mean SWE values for the period of MODIS record (2001-2014) are shown in Fig. 2b with a maximum between 32-35°S, 3000-4000 m ASL over the western slope of the Andes. The lack of important storms and the general increase in air temperature and solar radiation from spring to summer reduce the snow pack (Cornwell et al., 2016) that feeds the rivers in Central Chile (Cortés et al., 2011; Masiokas et al., 2006).

## 4 Drought identification and historical events

Because of the strong large-scale control of precipitation, year-to-year rainfall fluctuations exhibit a notable degree of spatial homogeneity in Central Chile. To track these common variations we utilize seven stations between 32°-37°S with complete annual rainfall records from 1915 onwards (Table 1), standardized by their corresponding climatological value (1970-2000 mean). The regional precipitation index (RPI, Fig. 3a) was then defined as the median of those seven accumulations for each year. The RPI has a correlation coefficient larger than 0.7 with individual station-based precipitation time series almost everywhere in Central Chile (Fig. 1d). Furthermore, RPI has no significant low-order correlation coefficient ($r_1 = 0.19$) and despite of positive skewness ($\gamma = +0.22$), it fits reasonably to a normal distribution.

Given our focus on annual or multi-year, regional-scale meteorological dry spells in Central Chile, an initial drought identification was obtained by considering those years when RPI was below 75%, equivalent to a 25% deficit in annual rainfall in Central Chile, a truncation level generally used for hydrological and agricultural applications (e.g., Sharma 1997; Bonaccorso et al., 2003). Such condition was observed in 24 years during the 1915-2009 period (about a fourth of the time) mostly as yearly or two-year droughts and some 3 three-year long events (Table 2). Consistent with the non-significant lag-1 autocorrelation of RPI, a chi-squared test reveals that drought spells evolve randomly in central Chile.

Considering the differences in the amount of rainfall variability (Fig. 1d), however, the simple drought selection based on RPI may favor the identification of dry conditions in the northern sector, calling for the use of the standardized precipitation index (SPI, McKee et al., 1993; Nuñez et al., 2014). To avoid bias in the computation of SPI we followed Stagge et al., (2015) and used the two-parameter Gamma distribution in combination with the unbiased Probability Weighted Moments to estimate its parameters (Hosking and Wallis, 1995). To better compare the severity of the 2010-2015 drought with the historical period, only the 40-year period 1961-2000 was used for parameter fitting (This period suffices for the calibration procedure; we tested longer periods in selected stations without significant changes in the final SPI computation). We employ the SPI calculated with a 12-month timescale ($SPI_{12}$) because this integrates fluctuations of natural (i.e., snow-accumulation) and man-made reservoirs that exhibit marked yearly cycles. Furthermore, we use the $SPI_{12}$ evaluated in December of each year ($SPI_{12D}$) that tracks the rainfall anomalies within the hydrological year (recall the austral wintertime precipitation maximum). As an example, Supplementary Fig. S1 shows $SPI_{12D}$ for three stations representative of northern (Ovalle, 31°S), central (Santiago, 33.5°S) and southern (Concepción, 37.5°S) Chile.

In each station, dry years are now identified as those in which $SPI_{12D} \leq -1$, a threshold that envelops moderate, severe and extreme droughts (McKee et al., 1993). The number of stations experiencing drought was calculated for each year since 1960 onwards considering 153 stations with complete $SPI_{12D}$ series between 32.5-36.5°S. The resulting time series (Fig. 3a) has a bi-modal distribution: one group of years with few stations in drought and the other with the majority of the stations afflicted by dry conditions, lending support to the concept of regional drought along Central Chile previously identified with the RPI.

Table 2 includes the main characteristic of dry events along Central Chile. Almost all the dry years selected with RPI $\leq 0.75$ feature more than 50% of the stations in the core of Central Chile having $SPI_{12D} \leq -1$. The intensity, duration and spatial extent of each drought were variable, as illustrated in Fig. 4 for selected cases. For multi-year droughts, intensity is defined as the average $SPI_{12D}$ while the magnitude is the sum of $SPI_{12D}$ (equivalent to drought severity in hydrological applications). Typically, the intensity decreases and the magnitude increases for long-lasting events (e.g., Sharma 1997). The persistent drought from 1967 to 1969 was severe from 30°-35°S. Another 3-year dry period occurred between 1994 and 1996, but this drought was more localized, less intense and had lower magnitude than the 1967-1969 event. The drought of 1998 was particularly intense and generalized, with $SPI_{12D} \leq -1.6$ (annual rainfall deficits larger than 40%) along much of Central Chile, while in 1962 dry conditions were more pronounced in the southern half of this region.

## 5 The recent megadrought (2010- 2015)

The Central Chile megadrought (MD) is readily evident in the time series of the regional precipitation index (Fig. 3a) as the uninterrupted sequence of dry years since 2010 with RPI ranging between 55% and 75%, that coincide with a regime shift identified by a Rodionov test at significant level $p = 0.05$ (Rodionov et al., 2004). This ongoing 6-year[1] drought is substantially longer than any other event since 1915 and every year during the MD has had more than 55% of the stations in Central Chile with $SPI_{12D} \leq -1.0$ (Fig. 3a) and 85% of them with $SPI_{12D} \leq -0.3$ (a minimum threshold for drought) . Given its duration, the MD magnitude is larger and, most notably, its spatial extent reaches farther south than previous multi-year events (Figs. 4d, 5a). The rainfall anomalies have some spatial variability as illustrated by the station-based maps for individual years conforming the MD (Supplementary Fig. S2). To the north of 32°S positive anomalies during 2010 and 2015 were associated with cut-off lows crossing this arid region in early winter (Bozkurt et al., 2016). Precipitation in 2014 was also near normal –albeit mostly below average- to the south of 35°S. In contrast, 2012 and 2013 featured annual rainfall deficits larger than 30% across much of the region.

### 5.1 MD recurrence in the historical record

Since the long duration of the recent drought has raised the attention of many stakeholders –from local farmers to national water authorities– in this section we further quantify how atypical the MD is within the instrumental record. To do so, we examine multi-year droughts (RPI<75%) in a broader context using the empirical frequency distribution of the 1-, 3- and 6- year average of RPI (Fig. 3, bottom panels). For the 1-year case, the intensity of the driest year within the MD (2013, RPI=56%) was within the historical distribution (1915-2009, light-blue bars) of RPI and much less intense than 1924, 1968 and 1998 (Fig. 3b), resulting in a low return period of ~20 years (obtained as the average recurrence interval). For 3-year events we contrasted the intensity (average RPI) of the worst 3-year sequence during the MD (2011-12-13) against two distributions (Fig. 3c). The light-blue bars represent the historical distribution of the average RPI considering an overlapping 3-year sliding window from 1915 to 2007 (that is, 1915-16-17, 1916-17-18,…,2007-08-09). The blue thick line is the distribution obtained from 5000 three-year periods formed by randomly selecting three years between 1915-2009 (such as 1919-2003-1971). The intensity of the worst three MD years has moved closer to the distribution's left tail but with a value still below the 1967-69 event. We applied the same procedure for 6-year events, contrasting the intensity of the full MD period (2010-2015) against the historical and synthetic distributions (Fig. 3d). The intensity of the MD is unprecedented (outside the historical distribution) and hardly obtained by chance ($p<0.02$). Complementing this empirical approach, the return period of a 6-year drought (RPI≤75%), given the RPI time-domain structure, is about 200 years when using the method proposed by Sharma (1997) for multi-year drought assessment. Both the empirical and Sharma's analyses emphasizes the extraordinary persistence of the recent MD, although the deficit in individual years was moderate.

---

[1] Our complete records extend until 2015 but dry conditions have persisted in central-southern Chile until the time of writing this work adding another year to the MD.

The maps of MD-averaged rainfall deficit and $SPI_{12D}$ (Figs. 5a and 4d) reveal important variability within Central Chile. This intra-regional variability is quantified using a frequency distribution of $SPI_{12D}$ in the long-record stations of Ovalle, Santiago and Concepción (Supplementary Fig. S3). The return period of the driest year during the MD increases from around 10 years in Ovalle, to ~20 years in Santiago to over 30 years in Concepción. Likewise, the intensity of the MD in Ovalle is only slightly negative (because the MD period includes some wet years in northern Chile) but is very unusual in Santiago and extraordinary in Concepción. Other (shorter) records are in broad agreement with the previous results (Supplementary Fig. S4). Thus, while the MD had some of the more noticeable, adverse effects in the arid/semiarid part of the country (CR2 2015), its magnitude and continuity increases southward. For many stations between 35°-38°S the recent 6-year drought is unprecedented -in terms of its length and magnitude- considering the records during the 20th century (Supplementary Fig. S4).

## 5.2 A millennium perspective

Our tree-ring record provides the first millennial length (1000-2014 AD) precipitation reconstruction for Central Chile (Fig. 6a), and represents the only tree-ring based hydroclimate reconstruction of this length in the entire Southern Hemisphere. Several decadal to multidecadal dry periods are observed during the 11th, 13th, 16th, and 17th centuries. Nevertheless, the negative conditions since the early 20th century appears to be the driest period on record, preceded by a century long wet period. This transition is consistent with the precipitation-driven glacier mass-balance dynamics of the region (Masiokas et al., 2006), which exhibit a sustained glacier shrinking during the 20th century following a major advance during the 19th century (LeQuesne et al., 2009; Masiokas et al., 2009; 2016). Since the beginning of the 21st century the reconstruction does contain a sustained dry period ending in the MD.

To frame the recent MD in this millennial perspective, we obtained the frequency distribution of the running mean anomalies in blocks of 1, 2,...,10 years for the AD 1000-2014 period (Box-and-whisker-plots in the left panel of Fig. 6). The red dots indicate the corresponding lowest anomalies during the MD (2010-2014) and from 2014 back when the blocks are >5 years. For all windows, the most recent anomalies are located within the lowest 2% portion of the past millennium distribution.

We also identified droughts along the reconstruction and grouped them according to their length. As expected, the number of droughts decreases with the event duration approximately following a geometric law of probability. Considering severe droughts (mean anomaly of -0.9; blue circles in Fig. 6b) we found 137 single-year events but only two events of five consecutive years, one of them being the 2011-2014 (MD) period. If we relax the drought selection to a mean anomaly of -0.15 (green circles in Fig. 6b) the period 2005-2014 is one of the two 10-year dry spells in the millennium record. Thus, our tree-ring reconstruction indicates that precipitation during the last decade and MD period has been extraordinarily low and with extremely few possible analogs in the context of the last millennium.

### 5.3 Large scale climate conditions

The large-scale context in which central Chile droughts typically occur is illustrated in Fig. 7 by the composite anomaly maps of austral winter precipitation, 500 hPa geopotential height (Z500) and sea surface temperature (SST) for the historical drought events and the MD period. As in past events, the MD in central Chile is connected with dry conditions across subtropical east Pacific (Fig. 7a,d). Weaker dry anomalies are also observed over the tropical Pacific while wetter anomalies prevail over the south Pacific straddling the southern tip of the continent. The Z500 anomalies provide a dynamical perspective on the occurrence of central Chile droughts. Of particular relevance is a dipole of positive anomalies across the subtropical Pacific and negative anomalies at midlatitudes, driving tropospheric-deep easterly wind anomalies over the western coast of South America centered at 40°S, the southern fringe of central Chile. In this region there is good correspondence between mid-level zonal flow and precipitation anomalies (Garreaud et al., 2013), and easterly wind anomalies have been identified as a recurrent ingredient of regional dry conditions by Montecinos et al., (2011). The Z500 dipole is very prominent in the MD composite (Fig. 7f) and present every year since 2010 (not shown).

The SST anomaly field during the MD (Fig. 7e) also shares several features with its historical counterpart (Fig. 7b), namely the weak cold anomalies over most of the subtropical SE Pacific and the horse-shoe pattern of warm anomalies rooted in the maritime continent. A remarkable difference between the two composites is the lack of significant cold anomalies along the equatorial Pacific during the MD. The scatter plot between winter values of Niño3.4 and RPI anomalies (Fig. 8) summarizes the statistical association between ENSO and the rainfall anomalies in Central Chile (cold-dry, warm-wet) commented before. Considering a threshold of ±0.5°C of the winter mean Niño3.4 index for El Niño/La Niña classification, only 2010 qualified as La Niña, ENSO-neutral conditions prevailed from 2011 to 2014, and 2015 qualified as a strong El Niño. To assess the likelihood of such dry sequence we made 5000 random extractions of five ENSO-neutral years from the historical RPI time series (1915-2009). The probability of having a 5-year mean rainfall deficit >25% is less than 4% and the probability of having such deficit in each individual year (as the recent MD) is less than 0.5%. Thus, although dry winters under ENSO-neutral conditions are not uncommon, a 4-year drought chain is unlikely, set aside the occurrence of a fifth dry year during a strong El Niño event.

A complete examination of the atmospheric dynamics sustaining the MD in Central Chile is beyond the scope of this paper, but this short analysis suggests that ingredients other than tropical ocean forcing are playing a role. This is in line with Boisier et al., (2016) who estimate that as much as a quarter of the rainfall deficit during the MD is attributable to anthropogenic climate change, mediated by altered mid- to high-latitude circulation in the Southern Hemisphere.

## 6 Impacts on hydroclimate and vegetation

### 6.1 Seasonal snow pack

Although the MD was identified on the basis of low-land precipitation data, it did have an impact in the cryosphere along the subtropical Andes. Let us begin by considering the seasonal cycle of the area covered by snow in the upper Maipo river basin (similar results are found in other Andean basins). Highlighted in Fig. 2a is the snow coverage during the MD years (2010-2015), from which a more rapid decline of the snow pack and an early end of the snow season is evident. At the end of September, the MD-mean fraction covered by snow was $61\pm7\%$, much smaller than the past decade mean ($73\pm9\%$). Year-to-year variations in the mid-spring snow coverage in the upper Maipo basin closely follows the winter rainfall accumulation in Andean foothills ($r = +0.73$, see also Masiokas et al., 2016) so the reduced snow pack extent during MD is largely explained by the lack of precipitation although some marginal effects of warmer conditions (see section 6c) can't be ruled out. To our knowledge, a separation of the precipitation and temperature effects in reducing the snowpack in central Chile has not been performed yet.

To track the evolution of the SWE over the Andes we relied on the distributed snow water equivalent (SWE) reconstruction (section 2c). The difference between the MD (2010-2015) SWE average minus the full MODIS period average is shown in Fig. 2c. Negative anomalies are found across the whole domain, with the largest values between 30 and 34°S on the western side of the continental divide. Between 3500-5000 m ASL, the SWE decreased >200 mm during the MD relative to the full period mean (Fig. 2d), significant at the 95% level. This SWE reduction is about 30% of the mean, a value similar to the MD-average rainfall deficit in low-level stations in the same range of latitude (c.f. Figs. 2c and 5a), emphasizing the close relationship between these two variables. The SWE decrease during the MD also extended eastward of the continental divide thus affecting the water resources of the agricultural region in westernmost Argentina (Bianchi et al., 2016; Rivera et al., 2017). Given that the accumulation and melt of the snow pack occurs within the hydrological year, we didn't find any cumulative effect of the megadrought in the SWE along the Andes of Central Chile.

### 6.2 Surface hydrology

Given the relatively small size of the basins (typically ≤ 10.000 km2) across central Chile, the precipitation deficit and snowpack reduction led to a concomitant decline in the annual river discharges with MD-average streamflow anomalies ranging from 20% to 70% relative to the long-term mean values (Fig. 5b). The bulk of the annual mean streamflow reduction is caused by a significant decrease in the summer peak flow (e.g., Fig. 9d), when water availability is critical, and some rivers in the northern sector dried during the fall season of some years within the MD. The spring-summer flows during the MD were typically above the 85% exceedance level (that is, they belong to the 15% drier portion of the distribution) in the major rivers of the region (MOP 2015). The volume of water stored in different reservoirs and hydrological systems also dropped dramatically during the MD. For example, the water volume of La Paloma, one of the largest irrigation reservoirs in Central Chile, and the groundwater level of the Alfalfares well have been at their historical lows since 2012 (Fig. 9).

A cursory comparison between Figs. 5a-b indicates that at nearly collocated stations the streamflow deficit is larger than the rainfall deficit. To explore this, we selected 119 basins with no major reservoirs or large irrigated areas (unimpaired flow) and their mean annual precipitation was calculated by interpolating catchment polygons with the gridded precipitation dataset described in section 2. Basin averaged precipitation deficit and outlet streamflow deficit were calculated as the

difference between their MD-mean values and their long-term means (1979-2009), normalized by their long-term means (Fig. 10a). The streamflow anomalies are generally larger than the precipitation anomalies during the MD, in agreement with the effects of drought propagation through the hydrological cycle (Van Lanen et al., 2013; van Dijk et al., 2013; Van Loon et al., 2014). Such hydrological amplification is more marked in the arid part of Central Chile (to the north of 34°S) where the normalized streamflow deficit can be twice larger than the normalized rainfall deficit. To the south of 35°S most data points

are closer to the 1:1 line (Fig. 10a). Exogenous factors (such as higher temperatures and vegetation changes) may be behind the increased deficit in streamflow in Central Chile during the MD (Bloschl and Montanari, 2010).

To further describe the impact of the MD in surface hydrology, the rainfall-runoff relationship is shown for two basins located in northern (30.9°S) and southern (36.2°S) Central Chile (Figs. 10c,d). Runoff is the streamflow normalized by catchment area and each point is a pair of annual mean rainfall and runoff (both calculated for hydrological years, April-

March). To assess their relationships, we computed the annual runoff coefficient (RC) as the ratio of annual mean runoff to annual mean precipitation, for the historical period (until year 2009) and for the MD. The MD-average RC is very similar to the historical coefficient in the southern basin, but much lower in the northern one.The significance of these RC changes was estimated by randomly extracting 5000 samples of 5 years from the full record and calculating their mean RC. We then compared the RC deviation during the MD against the synthetic distribution. Considering a threshold of 5%, the MD-change

in RC within the northern basin is significant while the change in the southern basin is not.

Mean runoff coefficients for the historical and MD periods were then calculated for each of the 119 basins in central Chile and the results are summarized in Fig. 10b, (highlighting with red borders the RC changes that are significant at the 5% level). Note that the historical water production (represented by the annual RC) varies widely between 0 and 1, which is expected given the different runoff mechanisms and climatic conditions across the study region (see Fig. 1). Similar to the

example in the two basins before (Figs. 10c,d), the RC during the MD did not change significantly in basins located to the south of ~35°S but experienced a substantial decrease in the most arid basins to the north of 33°S (Fig. 9b). Such decrease in RC indicates that northern, arid basins became "less productive" during the MD compared to previous dry periods, suggesting changes in the runoff generation processes. Although preliminary, these results agree with findings in Australian basins (Saft et al., 2016), where drier, flatter and less forested catchments exhibited a significant down-shift in their rainfall-

runoff relationship during an intense, multi-year drought.

### 6.3 Warming contribution and changes in PET

Annual mean maximum temperatures have experienced a rise since the late 1970s along the inland valleys of Central Chile and the western slope of the subtropical Andes (Falvey and Garreaud, 2007; Vuille et al., 2015), leading to warm anomalies

between 0.5° and 1°C during the MD relative to the past 30 years (Fig. 5c). As a representative example of inland stations, the MD-average temperatures in Santiago, are the warmest since 1960 (Supplementary Fig. S5). In contrast, hardly discernible trends are observed in the annual mean minimum temperature (Fig. 5d), thus causing an increase in daily mean temperature and its diurnal range. Both conditions favor enhanced evaporation from water bodies, evapotranspiration from crops and natural vegetation and snow sublimation (e.g., Hargreaves and Samani 1982). In most continental land-masses (including the rest of South America) the minimum temperature increased at a faster rate than maximum temperature during the second half of the 20th century narrowing the diurnal range (e.g., Vose et al., 2005), although this stabilized in the later decades. Thus, the contemporaneous strong/weak increase in daytime/nighttime temperatures and augmented diurnal range along central Chile is rather unique, the cause of which are as yet unknown.

To gauge the possible contribution of the warming to the MD impacts we employ the MODIS-derived potential evapotranspiration (PET, section 2c) that closely tracks the atmospheric water demand. Between 30° and 43° the annual average PET ranges from 800 to 2500 mm (Fig. 11a). Relative differences of PET obtained for the MD period (all months, 2010-2014), as compared to historical period (2000-2009), show a general increase ranging from 0 to 10% (Fig. 11b). In Mediterranean ecosystems (30° to 37°S) we observe a mostly neutral effect near the coast but a PET increase of about 5% in central valleys and up to 10% at higher elevations. To the south of 37°S, there is a more uniform PET increase during the MD, often larger than 5%. In absolute terms (Fig. 11c) we found two main areas where PET increased more than 50 mm/year during the MD: the interior valleys of northern Chile (30-33°S) and to the south of 36°S, suggesting substantial water stress for vegetation in these areas.

## 6.4 Changes in vegetation productivity

Here we analyze the Enhanced Vegetation Index (EVI; section 2c), closely related to gross plant productivity, from September to January thus encompassing the growing season for the entire study region. We then calculate the seasonal EVI anomalies for the MD period (2010-2015) over Central Chile compared to the previous decade (Fig. 12a). During the MD there is a severe browning (negative EVI anomalies) over most of the semiarid region to the north of 33°S as well as along the coastal range to the south of 39°S. The browning in the north is substantial, up to -20% of the historical mean at the grid level, and coincides with the region of MD-averaged rainfall deficit in excess of 30% (Fig. 5a) and PET increase over 100 mm (Fig. 11c). Somewhat unexpected is the EVI anomaly field between 34°-37°S, where we found a mix of small changes, some patches of browning and some patches of greening (positive EVI anomalies) despite the widespread drought conditions. The browning to the south of 39°S is less obviously related to the MD, since that region experienced near-normal rainfall conditions (albeit below average), but coincides with a sector of marked increase in PET.

Figure 12 also includes the scatter plot between winter rainfall anomalies and spring-summer EVI values in a box covering the northern sector of Central Chile. Considering the full period (2001-2015), the rainfall anomalies explains more than 50% of the EVI variance ($r = +0.76$). Unfortunately, the period with MODIS EVI data is still too short to discern if there is a cumulative effect of the protracted rainfall deficit upon vegetation, but we note that EVI anomalies during the MD were

consistently below their counterparts during individual dry years prior to 2010. This is consistent with stem growth anomalies from *Nothofagus obliqua* trees in the Andean foothills at 35°S that show little change during the extremely dry years of 1998 or 2003 but a consistent decline from 2010 onwards (Corvalan et al., 2014).

The vegetation responses to droughts are complex (e.g., Van der Molen et al., 2012, Vicente-Serrano et al., 2013) ranging from slow growth and reduced vigor, to loss of biomass and eventually plant mortality. Here we explore the vegetation responses as a function of two leading factors: land-cover (LC, section 2c) type and annual rainfall deficit. For each $5\times5$ km$^2$ MODIS-EVI pixel we obtained the predominant LC class (Fig. 13a) as well as the MD-mean rainfall anomaly allowing a bi-variate stratification of the EVI anomalies (Fig. 13b). The marked browning towards the north mainly involves the drop in vigor of shrublands, which is the dominant LC in this semi-arid region. Regardless of their location, shrublands show a high degree of vulnerability to water shortage as the EVI negative anomalies increases rapidly with rainfall deficit. The greening in the Mediterranean region occurred over exotic forest plantations (Eucalyptus and Pinus), croplands and, to a lesser extent, shrublands. For fast growing exotic forest plantations, the greening might be due to their characteristic rapid increase in leaf area since the first years following planting until their harvest (le Maire et al., 2011) which encompass a short rotation period (12-20 years), and for croplands it might indicate the combined effect of irrigation and warmer air temperatures. These results suggest that neither forest plantations nor croplands in Central Chile exhibit significant vulnerability to the rainfall deficit in the range of values observed during the MD. Pasturelands and native forests are mainly located in the southern half of Central Chile, where the MD rainfall deficit has been less marked but PET has increased. Nonetheless, those LC types in areas with more than 20% rainfall deficit do show a substantial browning.

**7 Discussion**

In spite of the integrative approach of our analysis, gaps remain on the origin and impacts of the MD, and here we briefly describe the challenges ahead. Within the bio-physical context, an outstanding question is the evolution of the ground water during the MD. This resource has been traditionally used in northern Chile, where some information exists on the aquifers (e.g., Hoke et al., 2004, see also Fig. 9b), but anecdotal evidence suggests that usage of ground water has expanded substantially toward the south in recent years, where little information and monitoring is available. In section 6.4 we noted a lack of "browning" and even some "greening" of the vegetation over parts of central Chile during the MD that may be the result of intense use of groundwater. The key question is whether or not these resources are being used in a sustainable manner, an issue extensively studied during droughts in California (Scanlon et al., 2012) and Australia (Hughes et al., 2012). Linking the hydroclimate and biophysical effects of the MD to socio-economic impacts and responses is a major task, well beyond the scope of this paper. For instance, evaluation of the economic burden of the MD requires industry-specific economic information disaggregated in time and space, and proper econometric methods to isolate the climate signal from other factors (e.g. Ding et al., 2011; Howitt et al., 2014). Yet, we consider relevant taking a first step in describing some of these aspects. Although over 75% of the Chilean population is now living in major cities, the country's economy is still driven by the exploitation of natural resources, among which agriculture and timber represent 9% of the total GDP (US$

23480) and 60% of the electricity is generated by hydropower (Supplementary Fig. S6). Semi-structured interviews conducted in urban and rural settings during 2015 revealed that nearly all the population identified some impacts of the MD, such as dryness in the landscape and reduction of superficial water bodies (Aldunce et al., 2017). Notably, most of the interviewed placed the onset of the dry conditions around 2010, coincident with the beginning of the MD. City dwellers were

5   somewhat protected from detrimental impacts of the MD because potable water and electricity companies managed to maintain a normal supply even at the height of this event (SISS 2015). To do so, the water suppliers had to rely more heavily on underground resources (ANNDES, 2014) and electrical generation shifted toward thermal power plants (Supplementary Fig. S6). Whether or not these measures were cost-effective and sustainable remains to be investigated.

Rural communities faced a far worse scenario than cities, with shortages of water for human consumption and agriculture

10   (e.g., Aldunce et al., 2017), an asymmetry that has been reported elsewhere (e.g., Swain 2015). The distress caused by the MD was recognized by county, regional and national authorities, that took several measures to alleviate impacts. Indeed, Chile, as most Latin American countries, has some drought management policies but they are loosely tied to objective indicators, with no feedback mechanism to adjust the policies (Verbist et al., 2016). In particular, the Chilean Ministry of Public Works uses several drought indices to declare water shortage (Decree N°1674-2012) at the county or river-basin level

15   for a maximum of six months. During this period, the authority can intervene the surface water allocation system (that normally works on the basis of a privatized market of water rights; Hearne and Donoso, 2005) to favour human consumption or other critical demands. Figure 14 shows the number of water shortage decrees (per administrative region and year) revealing a mismatch with the observed rainfall deficit. Likewise, the Chilean Ministry of Agriculture use several indicators to identify droughts at the communal level (Decree N° 81-2009), but the final decision about the area that will receive the

20   financial aid is taken by regional policymakers, and that area does not necessarily coincide with the one defined by the drought indicators (Fig. 14). For instance, the year 2013 was the driest during the MD but has a low number of decrees and agriculture emergencies. Coincidently, another survey study by Aldunce et al., (2016), indicates that water shortage decrees are the worst evaluated drought relief measure by the rural population. During a drought, county officials can also ask for financial resources to the central government in order to pay for potable water delivery by trucks to rural communities.

25   Figure 14 also displays those expenditures, which, at a national level have increased sharply during the MD from US$ 2 Mill in 2010 to US$ 45 Mill in 2015 (preliminary expenditure for 2016 has reduced to US$ 43 Mill; ONEMI 2016). Disaggregation by administrative region shows that more resources have been spend in the Bio Bio region (ca. 37-38°S), even after normalization by rural population, where the MD rainfall deficit and MD return period are the largest, followed by expenditures in the northern (semi-arid) sector of central Chile.

30   Thus, the complex link from rainfall anomalies to societal impacts can't be overemphasized and calls for integrated monitoring and management systems. Ongoing efforts to monitor droughts in Chile include monthly reports by the National Weather Service (station based SPI at several time scales) and Chilean Water Directorate (station based river flow and reservoirs levels), as well as international efforts as the Climate Data Library, a web application that allow national information relevant to drought to be collected (Del Corral et al., 2012), the SPEI Global Drought Monitor (Vicente-Serrano

et al., 2010) and the Latin American Drought Atlas (Nuñez et al., 2016). As evidenced during the mega drought, however, many challenges remain to take advantage of these efforts to support timely decision making.

## 8 Concluding remarks and outlook

Beginning in 2010, annual precipitation deficits ranging between 55% and 75% have afflicted Central Chile (30°-38°S) lowlands, the adjacent Andes cordillera and even westernmost Argentina. While intense (rainfall deficit as large as 70%) but short-lived (1-2 years long) droughts are a recurrent ingredient in the Mediterranean-like climate of this region, the recent event has several distinct features supporting its denomination as the Central Chile megadrought (MD):

- It is the longest, continuous dry spell in the historical record (1915 onwards). When considering a synthetic distribution of 6-year rainfall anomalies, the probability of an MD-like event is less than 0.5% and its estimated return period is ~200 years. Furthermore, we found only one MD analogue in a tree-ring based millennial reconstruction of regional precipitation.

- The MD reached farther south (>38°S) than previous droughts during the second half 20th century (often restricted to the north of 35°S). While the MD had some of the more noticeable, adverse effects in the semi-arid region of Central Chile (30°-33°S), its magnitude and continuity increased southward. For many stations between 35°-38°S the MD duration and severity is unprecedented considering the records during the second half of 20th century.

- It coincided with a very warm decade (MD-average anomalies > 1°C with respect to the 1970-2000 mean) in the interior valleys of Central Chile and the subtropical Andes.

- It occurred under mostly ENSO neutral conditions (except for La Niña year in 2010 and the strong El Niño in 2015), in contrast with tropical Pacific cold conditions that often accompanied dry years during the 20th century. This suggests factors other than tropical SST in sustaining the MD (e.g., anthropogenic climate change, see Boisier et al., 2016).

The ~30% precipitation deficit found at low-level stations in Central Chile also affected the subtropical Andes snowpack, where we observe a more rapid shrinking of the snow area during spring and a ~30% reduction in snow water equivalent by late winter at elevations between 3000 and 4000 m ASL, compared with past-decade values. The rain and snow deficit led to a concomitant decline in annual mean river flow across the highly populated Central Chile as well as the agricultural region of Argentina adjacent to the Andes (e.g., Rivera et al., 2017). The drought propagation through the hydrological cycle resulted in the amplification of the MD-mean signal, especially in the arid part of Central Chile where the normalized streamflow deficit can be twice larger than the normalized rainfall deficit. In this region we also found a substantial decrease in annual runoff-rainfall coefficient. This suggests that northern, arid basins became "less productive" during the MD compared to previous dry periods, signalling changes in runoff generation processes.

Changes in vegetation productivity, inferred from the Enhanced Vegetation Index, during the 2010-2015 period were complex. MD-mean, growing-season EVI anomalies over natural vegetation exhibits a strong decrease in productivity,

contrasting with irrigated croplands and fast growing exotic forest plantations that exhibit little sensitivity to drought. The browning of the vegetation was particularly clear in the case of the semiarid shrublands that dominate the sector to the north of 33°S, and there we found a hint of cumulative effects of the protracted rainfall deficit upon vegetation. Areas of marked browning tend to coincide with areas of increased potential evapotranspiration during the MD.

Recent studies suggest that the MD was perceived by most of the population, but the impacts in rural areas was more direct and more detrimental than in urban settings (Aldunce et al., 2016; 2017). A cursory analysis of the responses given by the State, the private sector and civil society indicate that most of them are based on the assumption that the mega-drought is a transient event and weakly related to the actual rainfall deficit on a given region or period. Additionally, the great number of different government agencies having jurisdiction over water resources makes coordinated action extremely cumbersome
and slow, as well as sub-optimal in social and economic terms.

The moderate intensity, but protracted, spatially extensive and warm character of the recent MD is consistent with the climate change signal expected for western subtropical South America. The present study serve as a basis to the efforts underway to assess the MD effects on water resources availability, agriculture, timber production, fires risks, ecosystem dynamics, as well as social behaviours and responses. Understanding the nature and impacts of the current multiyear drought
may contribute to preparedness efforts to face the projected dry, warm regional climate scenarios.

*Acknowledgments.* This research emerged from the collaboration with many colleagues at the Center for Climate and Resilience Research (CR2, CONICYT/FONDAP/15110009) that also provided partial funding. We thank Drs. M. Werner (Editor), R. Trigo and M.E. Enenkel for constructive criticism and comments on the manuscript.

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

**Table 1.** Stations with annual mean rainfall from (at least) 1915 onwards, used for the calculation of the regional precipitation index (RPI). Sources: DGA (National Water Directorate) and DMC (National Weather Service).

| Station Name | Latitude (S) | Longitude (W) | Elevation (m ASL) | Source |
|---|---|---|---|---|
| Ovalle | 31°47' | 71°22' | 301 | DGA |
| La Ligua | 32°27' | 71°16 | 58 | DMC |
| Peñuelas | 33°08' | 71°33' | 360 | DGA |
| Santiago | 33°26' | 70°41' | 520 | DMC |
| Talca | 35°25' | 71°40' | 100 | DGA |
| Chillan | 36°34' | 72°02' | 124 | DMC |
| Concepción | 36°46' | 73°03' | 12 | DGA+DMC |

**Table 2.** Drought events in central Chile

| Drought event | Year | RPI (%) | % Stations SPI12D≤-1 | Comments |
|---|---|---|---|---|
| D1 | 1916 | 65.22 | N/A | |
| D1 | 1917 | 72.42 | N/A | |
| D2 | 1924 | 30.57 | N/A | Driest year |
| D3 | 1938 | 73.10 | N/A | |
| D4 | 1943 | 71.82 | N/A | |
| D5 | 1945 | 74.92 | N/A | |
| D5 | 1946 | 56.25 | N/A | |
| D6 | 1955 | 68.20 | N/A | |
| D7 | 1962 | 57.65 | 55 | |
| D8 | 1964 | 51.67 | 71 | |
| D9 | 1967 | 67.37 | 66 | |
| D9 | 1968 | 32.87 | 79 | Third driest |
| D9 | 1969 | 69.40 | 33 | |
| D10 | 1973 | 67.15 | 29 | |
| D11 | 1976 | 69.12 | 39 | |
| D12 | 1985 | 64.65 | 71 | |
| D13 | 1988 | 64.77 | 70 | |
| D14 | 1990 | 59.92 | 75 | |
| D15 | 1994 | 70.90 | 38 | |
| D15 | 1995 | 73.52 | 39 | |
| D15 | 1996 | 51.75 | 69 | |
| D17 | 1998 | 32.00 | 73 | Second driest |
| D18 | 2003 | 65.10 | 27 | |
| D19 | 2007 | 52.30 | 72 | |
| D20 | 2010 | 72.22 | 65 | Megadrought |
| D20 | 2011 | 65.25 | 66 | Megadrought |
| D20 | 2012 | 72.60 | 53 | Megadrought |
| D20 | 2013 | 56.20 | 73 | Megadrought |
| D20 | 2014 | 71.00 | 51 | Megadrought |
| D20 | 2015 | 75.00 | 54 | Megadrought |

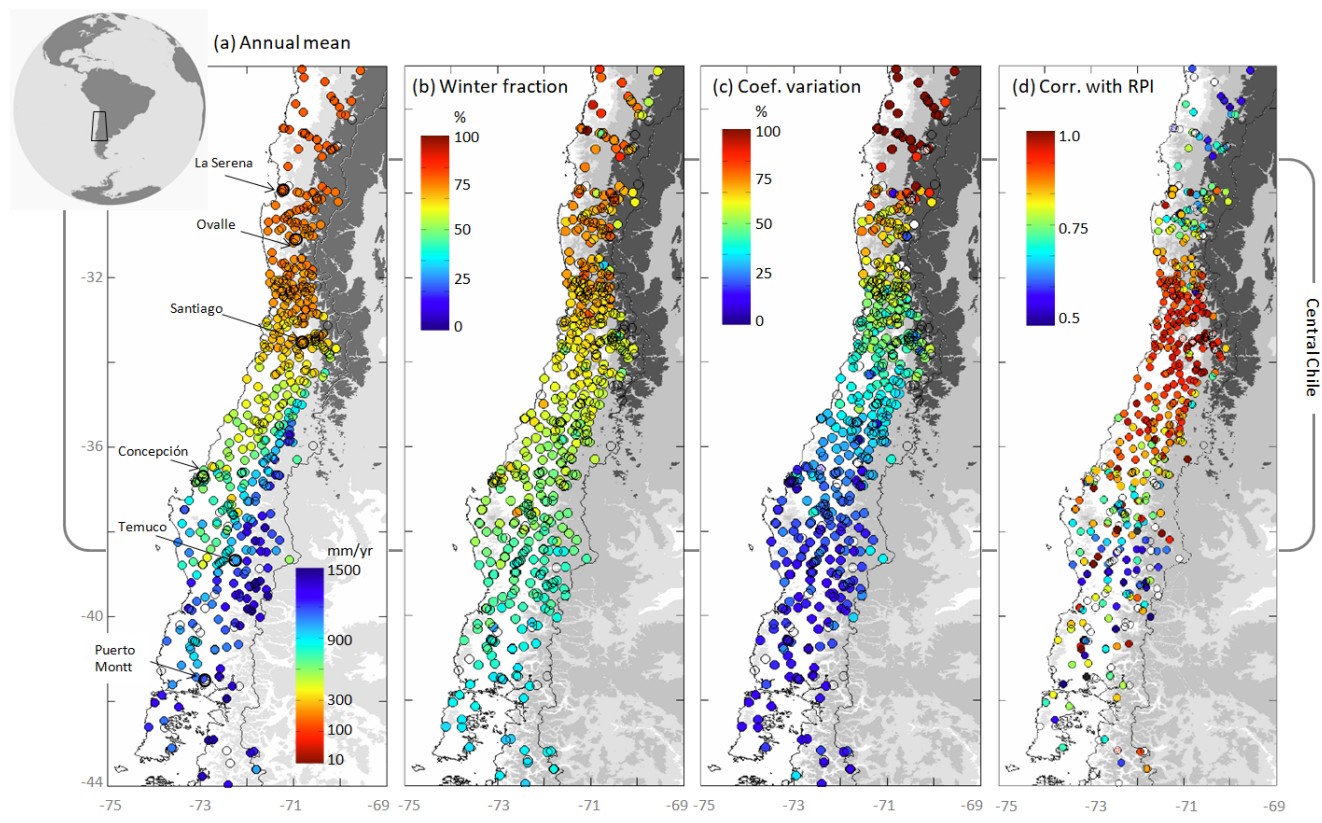

**Figure 1.** Long-term mean precipitation features along Central Chile. (a) Annual mean accumulation, (b) winter (MJJAS) fraction of the annual total, (c) coefficient of variation (interannual standard deviation divided by the long-term annual mean, in %) and (d) correlation with the regional precipitation index (annual time series). The variables are color-coded according to their value in each precipitation station. The solid lines are the coastline and the political border. Grey and black background areas indicate terrain elevation in excess of 1500 and 3000 m ASL, respectively. Reference period: 1980-2010.

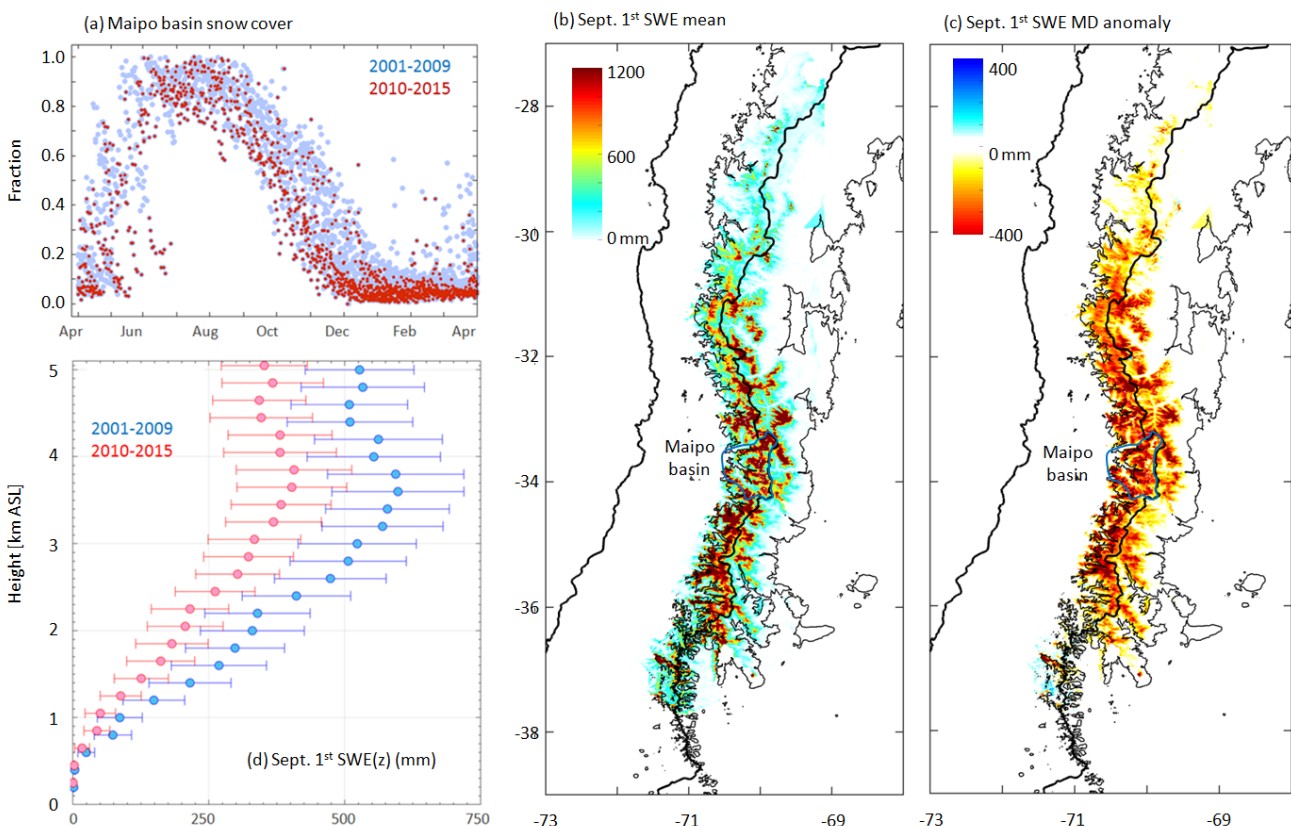

**Figure 2.** (a) Daily fraction of snow cover throughout the hydrological year in the upper Maipo river basin (33°-34°S; outlet at 900 m ASL; drainage area 5400 km$^2$) during 2001-2009 (blue circles) and the MD (2010-2015, red circles). Data from MODIS MOD10A1 product. (b) Late winter (September 1st) mean snow water equivalent (SWE) over the subtropical Andes. (c) September 1st MD-average (2010-2015) SWE minus last decade mean (2001-2009). (d) September 1st mean SWE in the region 32-35°S as function of the elevation during the MD period and the previous decade.

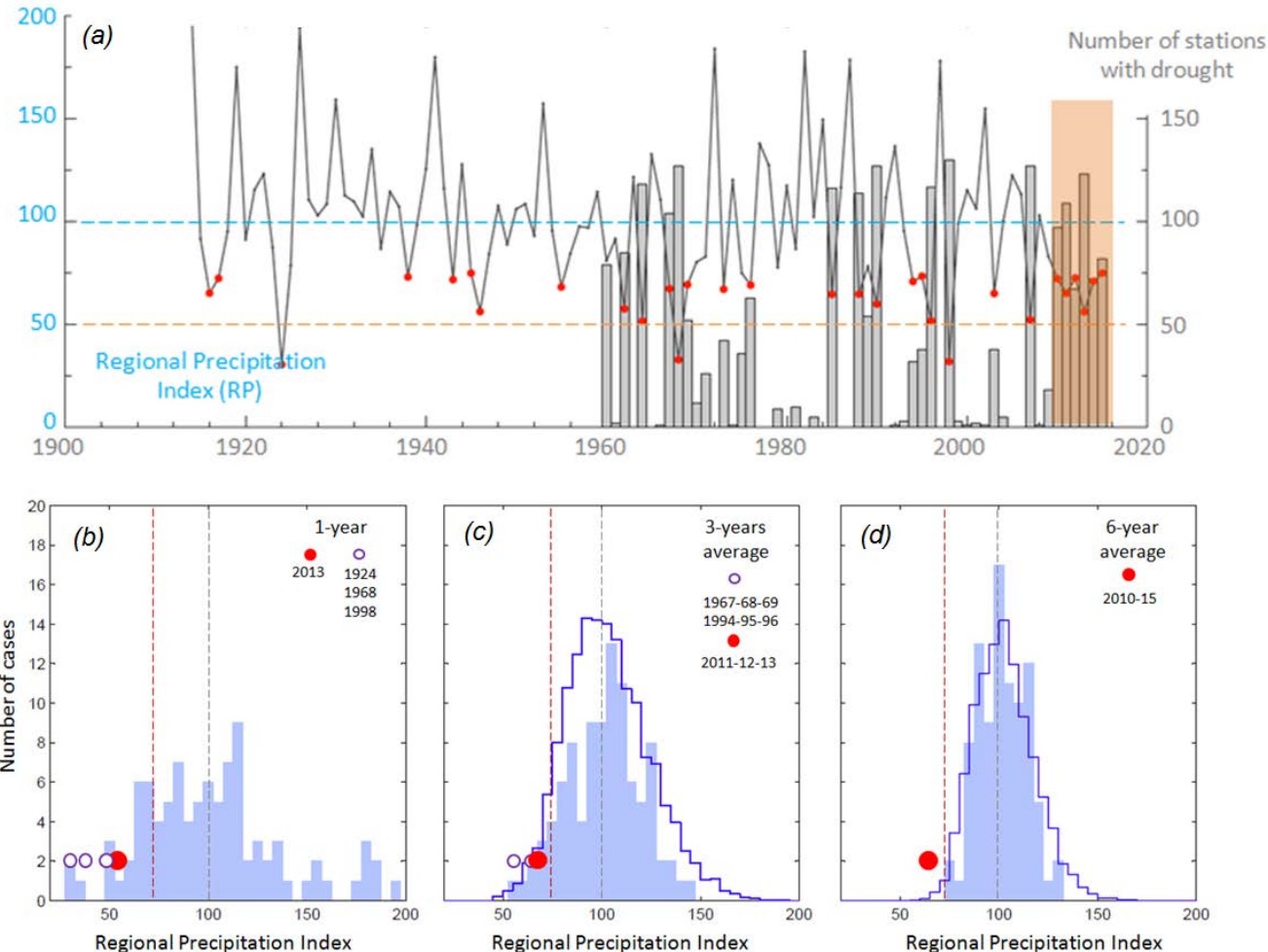

**Figure 3.** (a) Time series of the annual regional precipitation index (RPI, black line; red circles indicate drought years) and number of stations (bars, scale at right) in drought (SPI12-D<-1.1). The total number of stations in the core of Central Chile (32.5°-36.5°S) is 153. The orange area indicates the MD period. (b) Histograms of the historical RPI values (light blue bars; 1915-2009). The red circle indicates the worst year during the MD, the smaller purple circles indicate the values in contemporaneous droughts. (c) and (d) As (b) but for 3-years and 6-years average, respectively. The blue thick line is the distribution obtained from 5000 three-year periods formed by randomly selecting three/six years in the historical period.

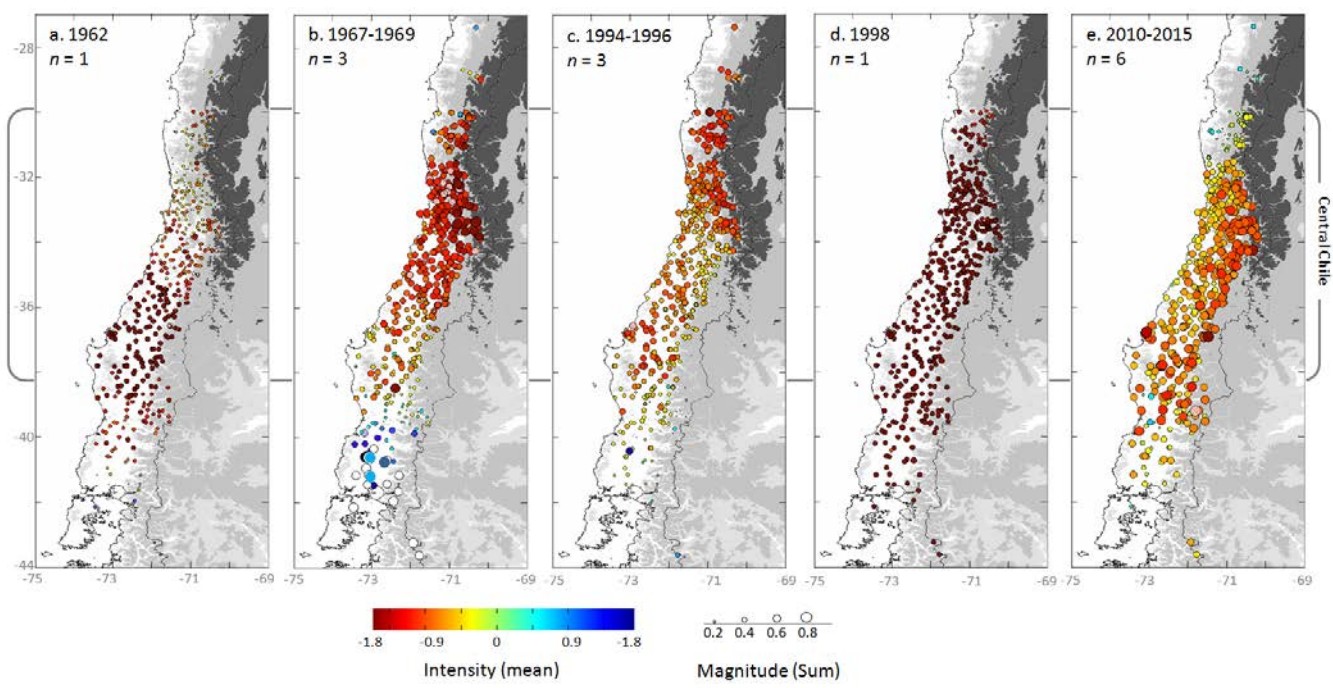

**Figure 4.** Station based, standardized precipitation index (December values of SPI12) for selected droughts. For multi-year droughts, the size of the symbol represents its magnitude (sum of SPI12-D) and the color its intensity (average of SPI-12D).

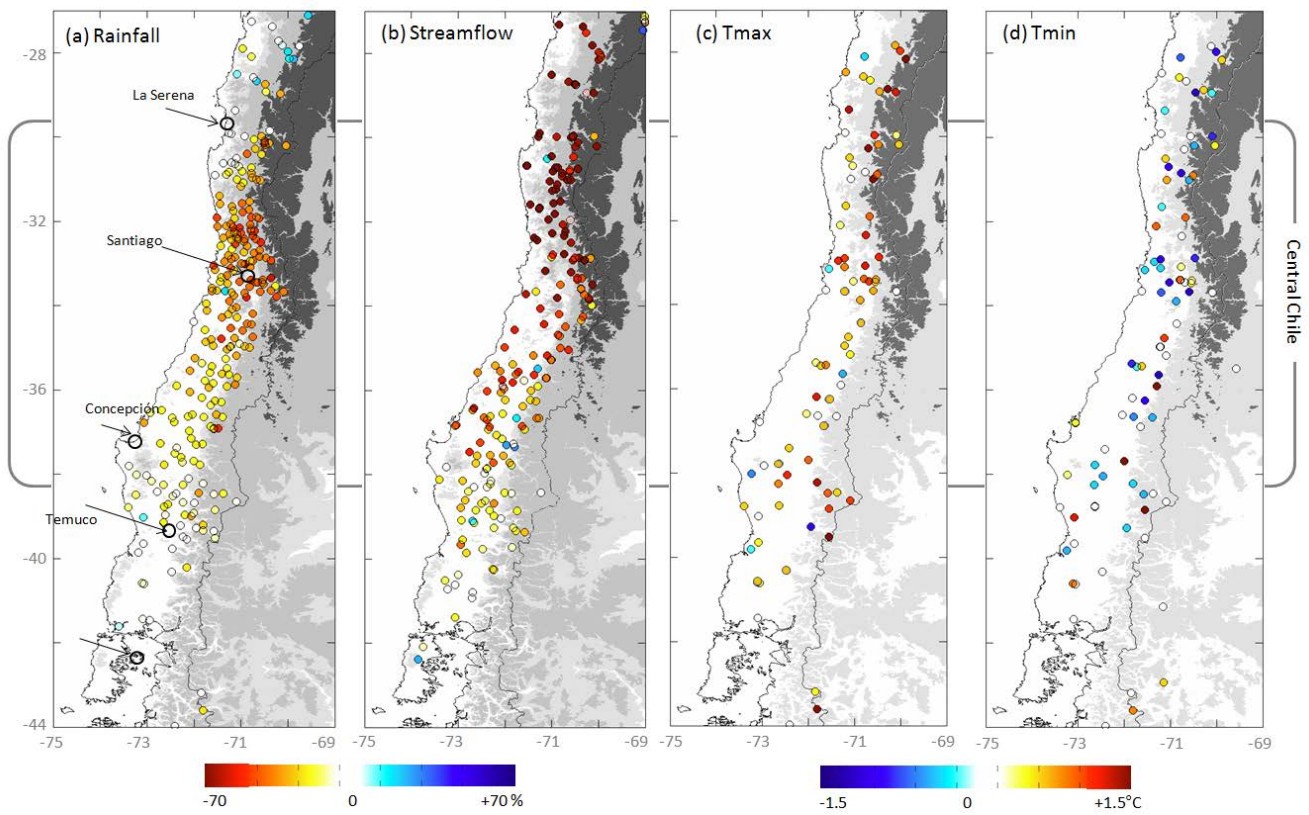

**Figure 5.** Station based anomalies during the Megadrought: (a) rainfall, (b) riverflow, (c) maximum temperature, (d) minimum temperature. In all cases we first calculate the annual mean (accumulation for rainfall), then average the 2010-2015 period and finally subtracts the respective long-term-mean (LTM; 1980-2010). Rainfall and riverflow MD anomalies expressed as a percentage of the LTM. Temperature anomalies are absolute values in °C. Note the existence of some outliers (e.g., warming station within a cooling region) probably due to problems in data quality.

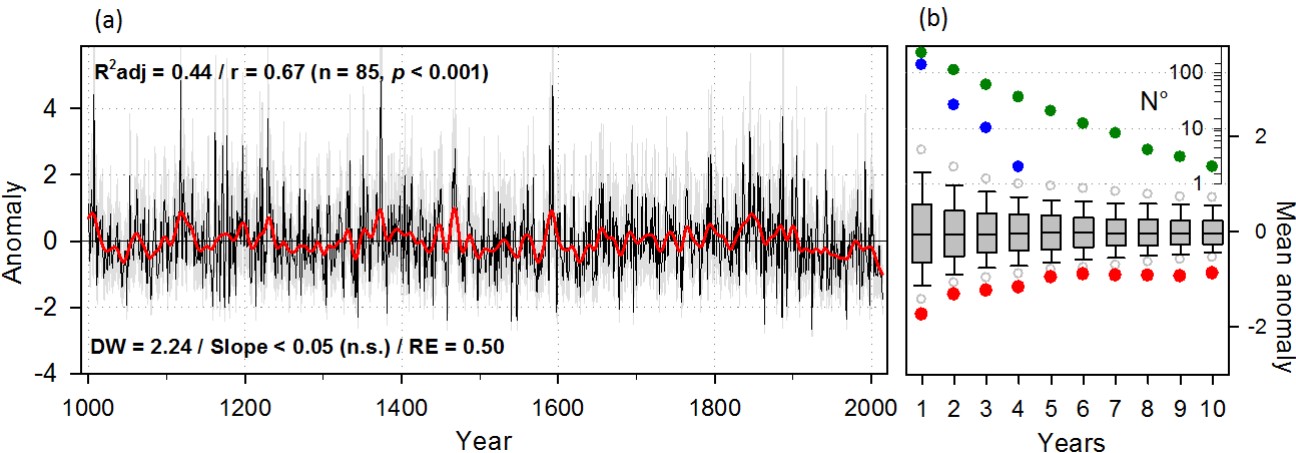

**Figure 6.** Left panel: The tree-ring reconstruction of winter-early summer (Jun-Dec) normalized precipitation anomalies from Central Chile plotted annually from AD 1000 to 2014. The red line indicates a 25-year Gaussian filter to emphasize the multi-decadal variations. The shaded area denotes the 1±root-mean-square error from the residual fit to the instrumental series. Right panel bottom: Box-and-whisker-plots (gray and black) indicating the median, the 5th, 10th, 25th, 75th, 90th and 95th percentiles of the running mean (RM) anomalies of the reconstruction from 1 to 10 adjacent years. The red dots indicate the lowest mean values of RM from the 2010-2014 reconstruction period for the case of RM < 5 (years), and for the case of RM > 5 (years) the corresponding RM starts from 2014 back. Right panel top: Number (N°) of droughts (log scale) in the reconstruction with a threshold of -0.15 (green dots) and -0.9 (blue dots), in relation to the years duration.

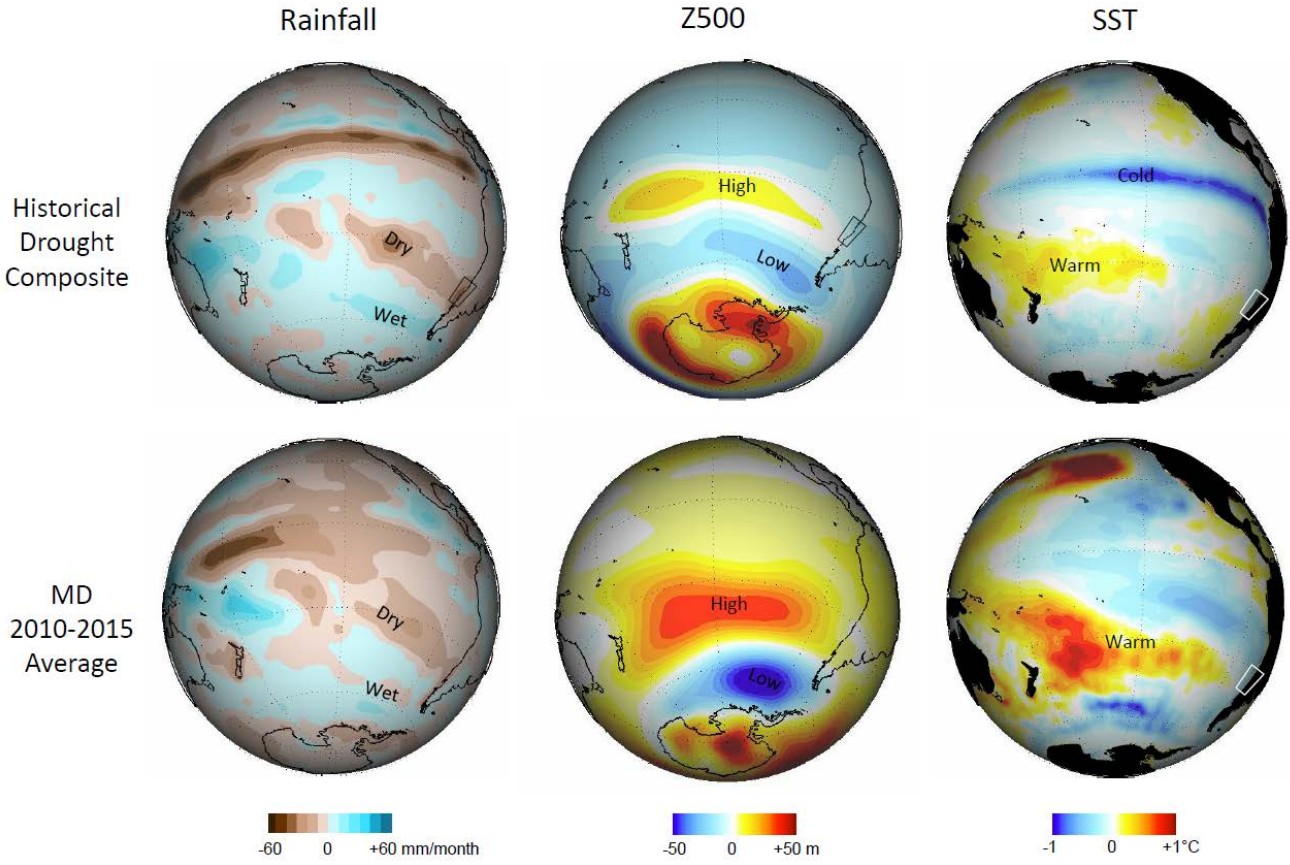

**Figure 7.** Composite anomaly (departure from long-term mean) maps of austral winter (MJJAS) precipitation (from the Global Precipitation Climatology Project available since 1979; Adler et al. 2003), sea surface temperature (SST, from NOAA extended-reconstructed sea surface temperature available since 1860; Smith et al. 2008) and 500 hPa geopotential height (Z500, from NCEP-NCAR reanalysis available since 1948; Kalnay et al. 1996) during droughts in central Chile. Upper panels: historical droughts according to Table 2 (1948-2009). Lower panels: megadrought period (2010-2015).

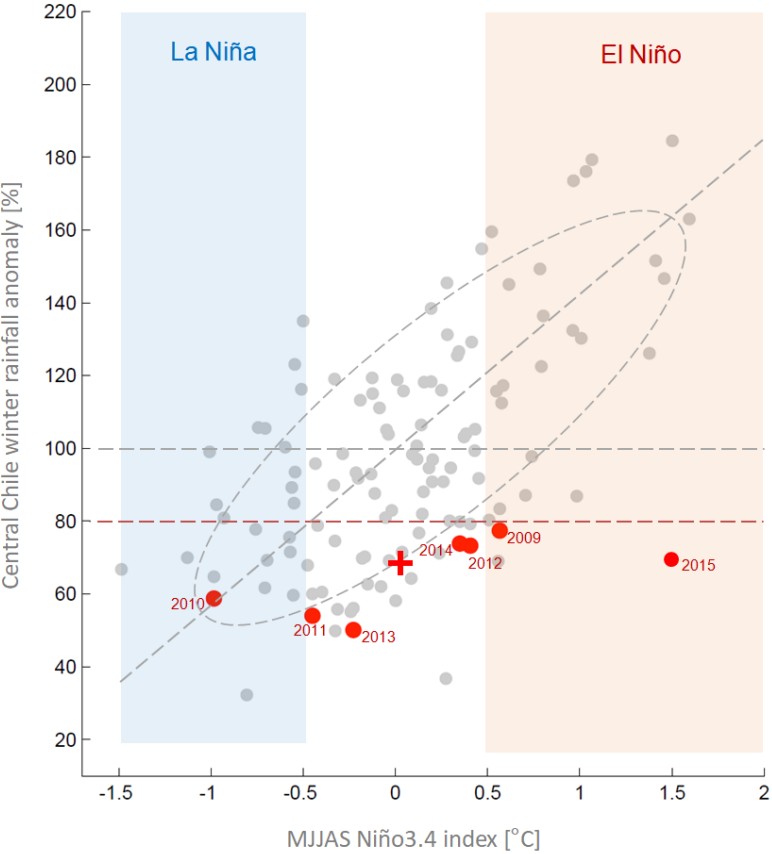

**Figure 8.** Scatter plot between winter (MJJAS) average of Niño3.4 and Central Chile precipitation index (RPI). Data from 1915 onwards; the years forming the MD are highlighted in red. The Niño3.4 index is the area average SST anomaly in the region 5°S-5°N and 170°-120°W.

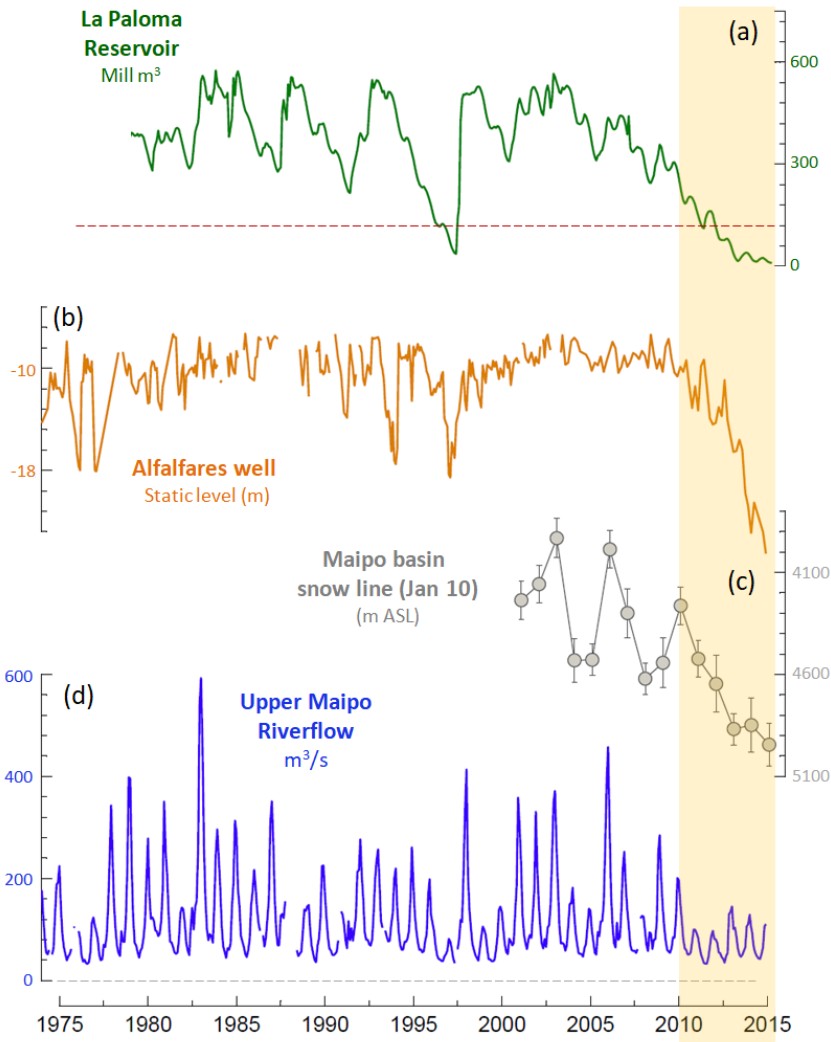

**Figure 9.** Selected hydrological impacts of the MD (highlighted by the yellow area). (a) Green curve: Monthly mean water volume in the La Paloma reservoir (32°S). (b) Orange curve: Monthly mean water depth in the Alfalfares observation well (32.5°S). (c) Grey curve: January 10th (early summer) height of the snowline between 33-34S (note that the scale has been reversed). (d) Blue curve: Monthly mean Maipo river flow as measures in station El Manzano (850 m ASL, 33.5°S).

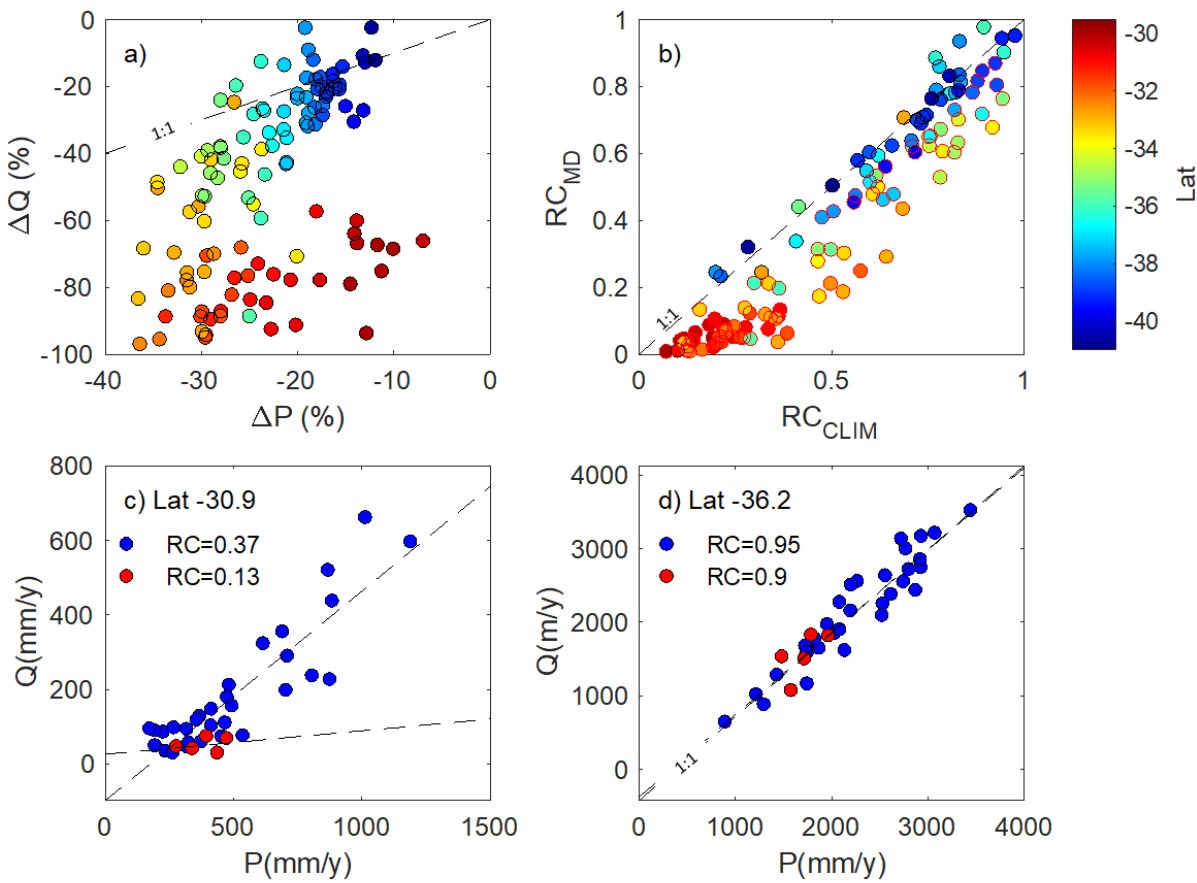

**Figure 10.** (a) Scatter plot between the meteorological and hydrological MD-standarized deficits (DP and DQ, respectively) in 119 basins along central Chile. For both variables the deficit was calculated as the difference between the MD-average (2010-2015) values minus the long term mean (LTM, 1979-2009) and then divided by the LTM. (b) Scatter plot of the annual runoff coefficient (RC) calculated for the historical period (1979-2009) and the MD period (2010-2015) in 119 basins along central Chile. The basins that experienced a significant change in RC are marked with red borders. In panels (a) and (b) the color of the circles indicate the latitude of the basin outlet. (c) Scatter plot between the area averaged annual mean precipitation and the basin runoff (annual mean river flow divided by the basin area) for rio Grande at Cuyano basin (30.9°S). (d) As (c) but for Longavi at Quiriquina basin (36.2°S). In panels (c) and (d) the rainfall-runoff values during the years forming the MD are indicated in red. In panels (a), (b) and (d) the dashed line is the 1:1 line. In panel (c) the dashed lines are regression lines.

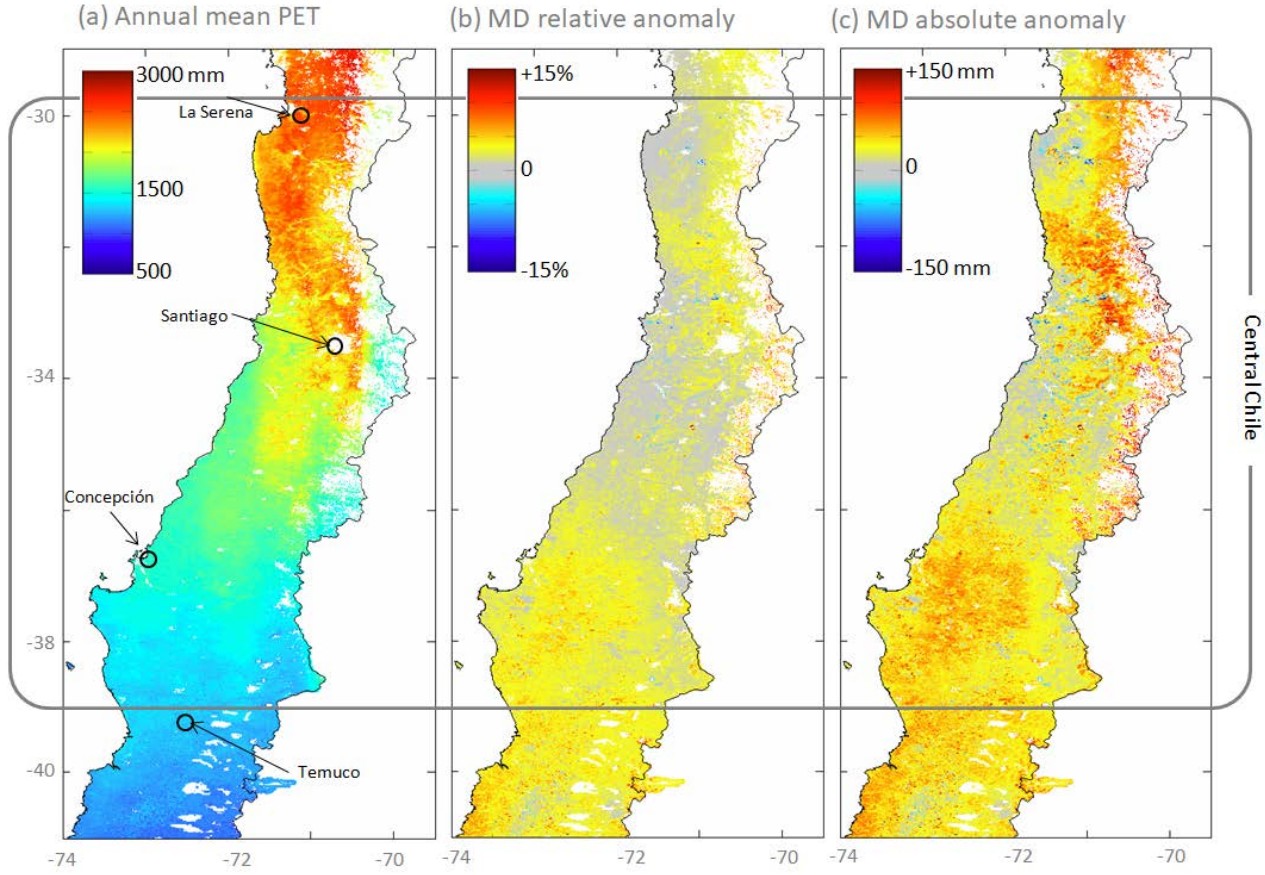

**Figure 11.** (a) Annual mean MODIS-derived potential evapotranspiration (PET) over Central Chile (2001-2015). (b) MD-averaged PET anomalies relative to the long-term mean. (c) MD-averaged PET absolute anomalies.

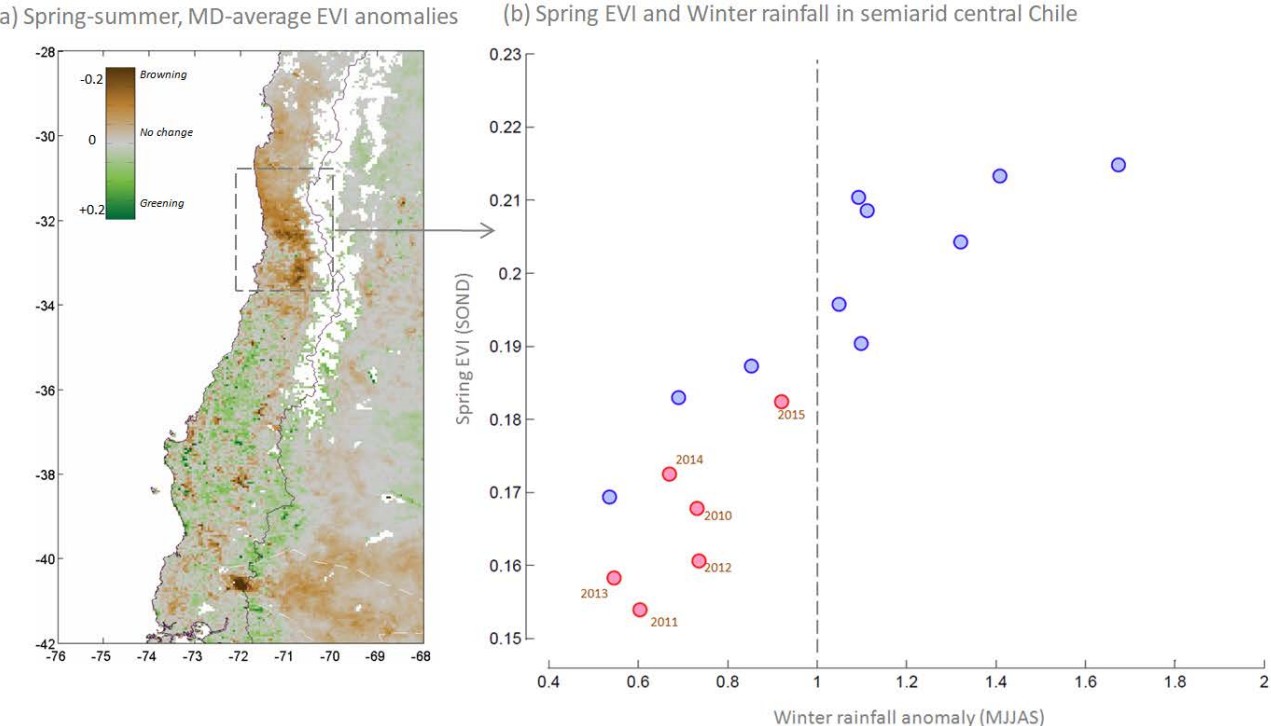

10    **Figure 12.** (a) Spring-summer (SONDJ) EVI anomalies during the MD (difference between average 2010-2015 minus average 2001-2009). White areas indicate no or too little vegetation. (b) Scatter plot between the winter (MJJAS) gridded rainfall anomalies (as the fraction of the long-term mean) and the spring (SOND) MODIS-derived enhanced vegetation index (EVI) area averaged in a box over northern Central Chile (see panel a). Data from 2001 to 2016. The years conforming the MD are highlighted in red.

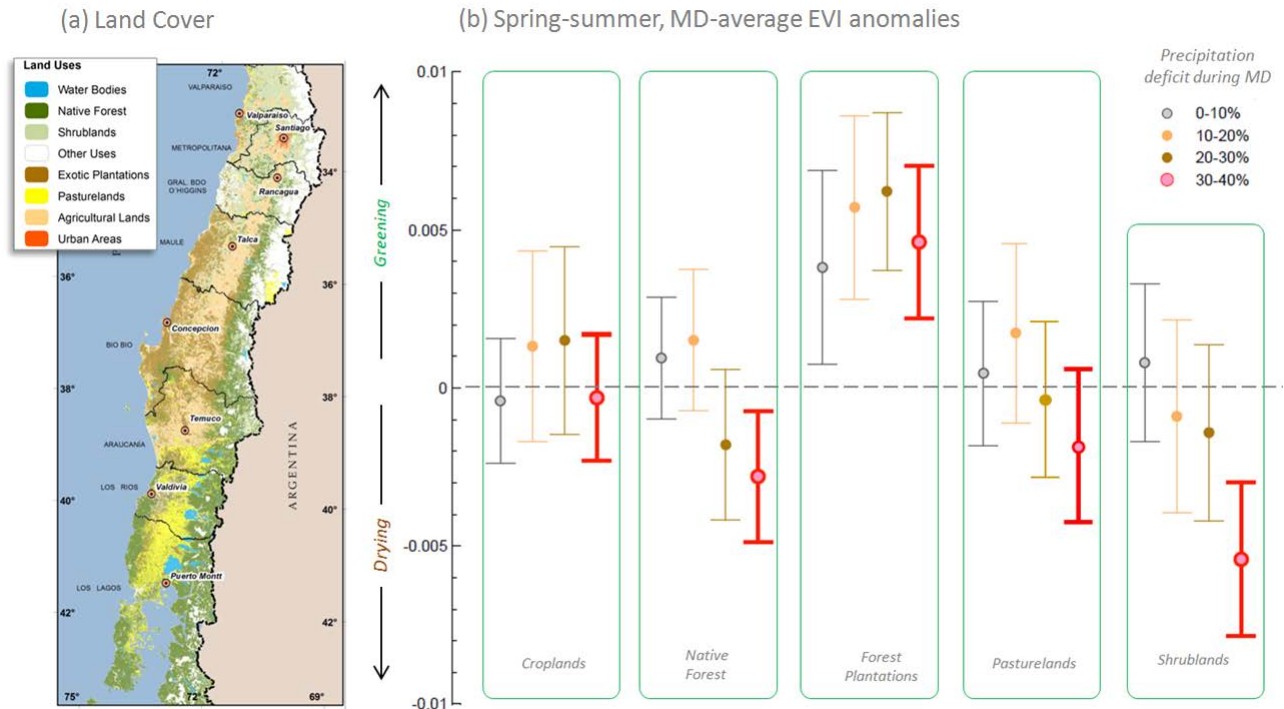

**Figure 13.** (a) Land cover (LC) classes for 2014 over Central Chile. (b) EVI changes during the MD period (2010-2015) with respect to the previous decade (2001-2009) stratified according to the LC and the rainfall deficit during the MD. There are many 5×5 km$^2$ pixels for each LC and rainfall deficit categories: the circles indicate the median change, the errorbars are ±1 standard deviation.

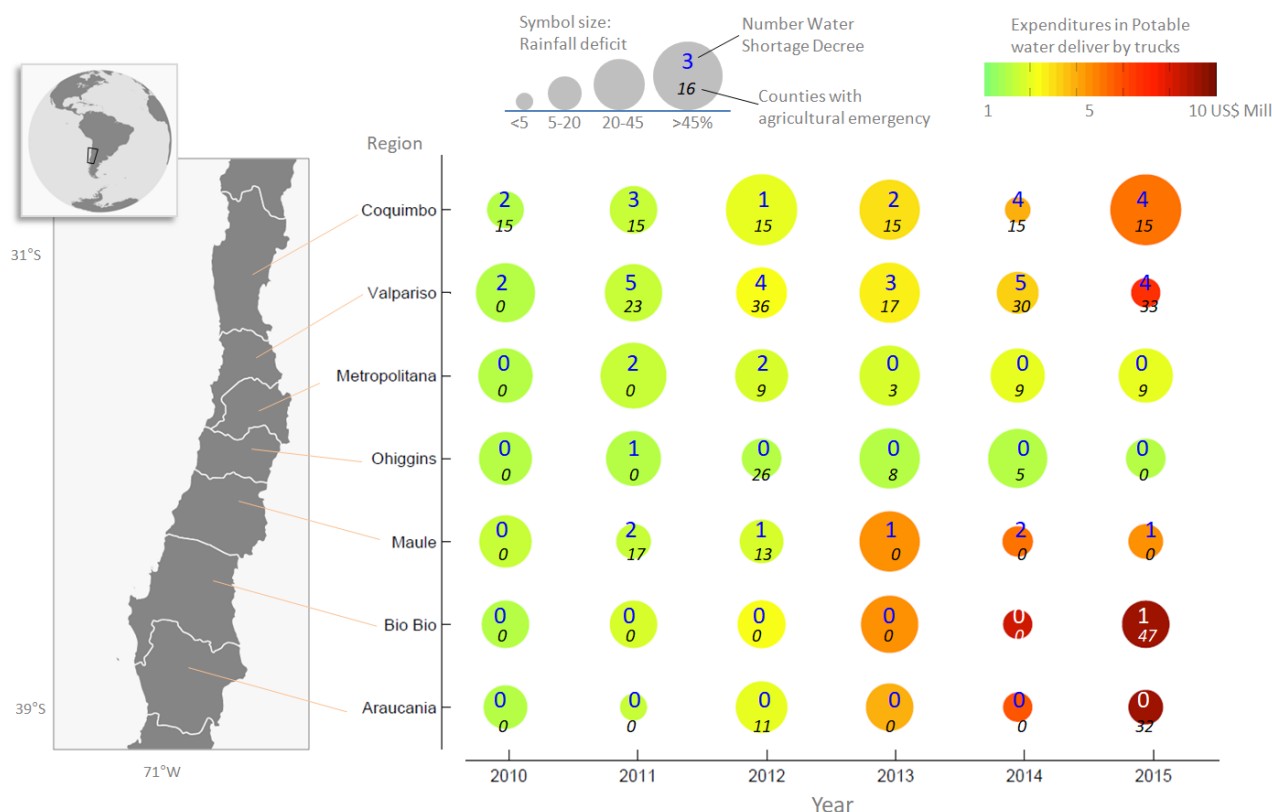

**Figure 14.** Evolution of the meteorological drought and the state response. The data has been disaggregated by year and administrative region (see map to left). The circle size is proportional to the rainfall deficit, and the circle are colored according to the amount of money spent to deliver potable water by trucks to rural communities, according to the scales atop.
10   The blue, larger numbers indicate the number of water shortage decreed by the Ministry of Public Work (MOP). The smaller numbers in italic indicate the number of counties in agriculture emergency as determined by the Ministry of Agriculture (MINAGRI).

