# Peer review of "The 2010-2015 mega drought in Central Chile: Impacts on regional hydroclimate and vegetation"

_Hydrology and Earth System Sciences, 2017_

## Referee Comment (RC1) · R. Trigo (Referee) · 9 Jun 2017

**Review**

**"The 2010-2015 mega drought in Central Chile: Impacts on regional hydroclimate and vegetation"**

by

René Garreaud, Camila Alvarez-Garreton, Jonathan Barichivich, Juan Pablo Boisier, Duncan Christie, Mauricio Galleguillos, Carlos LeQuesne, James McPhee, Mauricio Bigiarini

This manuscript focus on the prolonged and intense Drought episode that struck Chile between 2010 and 2015. The authors provide a comprehensive characterization of this so called Mega-Drought event from various perspectives (meteorological, hydrological, anomalous climate dynamics, vegetation dynamics) but also considering the long-term context (last millennium), and finally providing some framing within regional warming background. The work is very interesting to read, with plenty of informative figures. The major problems I see are related with the level of novelty of this manuscript taking into account the contents of two other papers by the authors with some overlap in contents. I'm also particularly concerned with the amateurish attitude of the authors in relation to citations with so many missing and wrong references. Thus, I believe the paper can be accepted if the authors improve the manuscript taking into consideration the following clarifications listed below.

**Major issues**

**1. Level of originality**

Despite the overall good quality of the work presented here I must confess that an interested reader cannot be entirely sure on the level of originality of the contents included in the manuscript, particularly taking into account the two sister publications carry out by these authors, and covering (at least in part) the same Mega-drought event (Boisier et al., 2016 and Garreaud et al., submitted). While I can understand perfectly well that such a major event can be characterized from multiple perspectives, it is not entirely clear the level of superposition (if any) among these three manuscripts. Please provide a clarification on this important issue.

**2. References**

The authors were extremely careless with the references. It is unacceptable that you have so many missing and wrong references, including papers by the authors (?). This is quite distracting when a reader is trying to put the scientific questions in context of previous literature. Without guarantying that I've cover all the problems please check the following:

a) The following references are missing. Please add in the final reference list:

- (Page 3): Obasi et al (1994)

- (Page 3): Hao et al (2014)
- (Page 3): Gleick (2015)
- (Page 3): Cooley et al. (2015)
- (Page 3): Cook et al. (2015)
- (Page 3): Masiokas et al (2016) (could be Masiokas et al (2006) ?)
- (Page 4): Fuenzakida et al (2007)
- (Page 4): MOP  (2013)
- (Page 4): Garreaud et al. (2017)
- (Page 5): Boisier et al. (2017)
- (Page 5): Cook and Kairiukštis (1990)
- (Page 5): Michaelsen (1987)
- (Page 6): Vicente-Serrano et al (2010)
- (Page 6): Schut et al. (2015)
- (Page 6): Miller (1976)
- (Page 7): Rivera et al. (2001)
- (Page 10): Jones et al. (2009)
- (Page 11): Montecinos et al. (2011)
- (Page 12): Van Lanen et al. (2013)
- (Page 12): Van Loon et al. (2014)
- (Page 12): Bloschl and Montanari (2010)
- (Page 13): Hargreaves and Samani (1982)

b) The following papers are listed in the final reference list but not cited in the manuscript. Are these relevant? If the answer is positive than cite them in the manuscript, where appropriate.

- Bréda et al. (2006)
- Chaves et al (2003)
- Falvey and Garreaud (2009)
- Hatchett et al. (2015)
- Hernández et al. (2016)
- Ji and Peters. (2003)
- Naresh et al. (2009)
- Norte chico, Chile ????
- Shukla et al. (2015)
- Tabari et al. (2012)
- Wang et al. (2014)
- Wu et al. (2007)

**Minor suggestions**

**1. (Introduction, 2ⁿᵈ parag)** The authors introduce several major droughts that have occurred around the world in the last two decades. Taking into account the amount of people affected and the outstanding implications I would like to suggest to add the Middle East (or Fertile Crescent) drought between 2007-2012 (Trigo et al., 2010; Kelly et al., 2005)

Trigo R.M., Gouveia C., Barriopedro D., (2010) " The intense 2007-2009 drought in the Fertile Crescent: Impacts and associated atmospheric circulation", Agricultural and Forest Meteorology, 150, 1245-1257

Kelley C.P., M. Shahrzad, M.A. Cane, R. Seager, Y. Kushnir (2015) "Climate change in the Fertile Crescent and implications of the recent Syrian drought". Proc. Natl. Acad. Sci., 112 (11) (2015), pp. 3241–3246

**2. (Page 5, lines 22-23)** This statement that winter precipitation corresponds to >75 needs a reference to support it. More importantly, this is not consistent with Fig. 1b where winter precipitation for regions south of 33ºS represents less than 75%. Please rewrite sentence, adapting to the large N-S gradient of winter precipitation contribution.

**3 (Page 6, lines 32-33)** The marked West-East gradient in precipitation is not so clear at all latitudes as it happens mostly in the central section of Chile. North of 33ºS and South of 38ºS it appears to be negligible.

**4 (Page 7, lines 23)** I believe this low-order correlation refers to the autocorrelation coefficient? If so please clarify it.

**5 (Page 8, lines 2-3)** If you have continuous precipitation data from 1960 until 2015 why restricting the historical comparison period to the 1961-2000 (40 years) instead of considering 1960-2009 (50 years)?

**6 (Page 8, lines 3,5,9)** Although there are no standard procedures, the SPI acronym is usually employed as such, the temporal scales should be added as indices (or brackets). Please consider adapting the cumbersome SPI-12D to $SPI_{12D}$ or even $SPI_{12}$

**7 (Page 11, lines 27-28)** Can you provide some additional information or literature for the Andes regarding the separation of the role played by diminishing precipitation and increasing temperatures in terms of reduced snow pack.

**8 (Page 12, line 11)** Please check if Fig.8 should be Fig.9 here.

**9 (Page 13, line 5)** Please check if Fig.9b should be Fig.10b here.

**10 (Page 13, lines 18-22)** This increment of DTR is consistent with the remaining of South America? Until the AR4 IPCC report in 2007 the DTR was diminishing in most areas of the world, but that has changed in the last decade. Can you provide a little bit more of regional (remaining S. America) and temporal (changes in DTR trend signal) context.

**11 (Page 13, line 25)** Please check if Fig.10b should be Fig.11b here.

**12 (Page 13)** Please check carefully all Figure numbers. Some appear to be lagging by one (e.g. Figure10c means Figure11c, Figure12b means Figure13b).

**Figures**

**Fig.1d** The scale used is a bit misleading. Every station appears with the same reddish color and it is very difficult to distinguish regions. What is the point of presenting the range of possible values between -1 and 1? Please compress the possible values to the range [0 1] or even [0.5 1]. That will provide a much more informative plot.

**Fig.5a and Fig.5b**. Can you explain how come a few stations present a positive rainfall trend (Fig.5a) or streamflow trend (Fig.5b) in the midst of strong negative trends everywhere else?

**Fig.9** Please provide a clear link between each subpanel letter (a,b,c,d,e) and the corresponding section in the figure caption.

**Fig.10**. Please provide the meaning of the regression dashed lines in the figure caption.

**Fig.12**. If you describe first Fig.12b and then Fig.12a why not provide the information in the logical reverse order, i.e. place the map on the left (becoming Fig.12a) and the scatter plot on the right (becoming Fig.12b)?

Ricardo Trigo

---

## Referee Comment (RC2) · M.E. Enenkel (Referee) · 20 Jun 2017

Summary

The study of Garreaud et al. puts the recent perennial drought conditions in Chile into a multi-dimensional historical and thematic context. The strengths of the manuscript lie in the holistic approach, the level of analytical detail, the visualization of drought conditions on a regional/global scale and the link to long-term climate conditions via tree-ring analysis. I agree with my fellow reviewer that, in the context of existing literature about the Chilean MD (even from the author's themselves), the novelty of the manuscript is questionable. However, what bothers me more is that the paper implies

improved drought preparedness via an "understanding of the nature and biophysical impacts of the MD". Unfortunately, it fails to relate the presented findings to any kind of decision-making or socio-economic response. I see potential in the paper if the authors manage to link the tracking of the MD through the season(s) via different variables and the MD's historical/climatic context to any kind of suggestion for decision-support, such as socio-economic countermeasures (e.g. changes in agricultural practices, consideration of seasonal climate forecasts). I understand that an in-depth consideration of the manuscript's findings related to decision-support or climate-change adaptation is out of scope, but even a superficial discussion would improve the manuscript's

For a detailed review please have a look at the attached .pdf document.

[Figure]

Review of:

"The 2010-2015 mega drought in Central Chile: Impacts on
regional hydroclimate and vegetation"

René Garreaud, Camila Alvarez-Garreton, Jonathan Barichivich, Juan Pablo
Boisier, Duncan Christie, Mauricio Galleguillos, Carlos LeQuesne, James
McPhee, Mauricio Bigiarini

**Summary**

The study of Garreaud et al. puts the recent perennial drought conditions in
Chile into a multi-dimensional historical and thematic context. The strengths of
the manuscript lie in the holistic approach, the level of analytical detail, the
visualization of drought conditions on a regional/global scale and the link to
long-term climate conditions via tree-ring analysis. I agree with my fellow
reviewer that, in the context of existing literature about the Chilean MD (even
from the author's themselves), the novelty of the manuscript is questionable.
However, what bothers me more is that the paper implies improved drought
preparedness via an "understanding of the nature and biophysical impacts of
the MD". Unfortunately, it fails to relate the presented findings to any kind of
decision-making or socio-economic response. I see potential in the paper if
the authors manage to link the tracking of the MD through the season(s) via
different variables and the MD's historical/climatic context to any kind of
suggestion for decision-support, such as socio-economic countermeasures
(e.g. changes in agricultural practices, consideration of seasonal climate
forecasts). I understand that an in-depth consideration of the manuscript's
findings related to decision-support or climate-change adaptation is out of
scope, but even a superficial discussion would improve the manuscript's
overall relevance.

**Main comments**

- The manuscript needs to be more clearly distinguished from the
authors' other publications
- References and figures need to be reviewed (some are missing, some
only mentioned in the reference list, but not in the text)
- Although 2010 was the strongest La Niña year during the MD period
2011, 2013 and 2014 were La Niña years as well. Could these
conditions have contributed to the MD's persistence?

**General comments**

- Whenever you talk about rainfall deficits and related percentages
please mention the reference period
- Instead of describing selected drought events in the introduction I
suggest you provide more general statistics, if available (e.g. from
UNISDR; EMDAT might not be a good choice)

[Figure]

**Fig. 1.**

---

## Author Comment (AC1) · 15 Jul 2017

Original comments in quotes "..."

Reviewer 1 (Dr. Trigo) Recommendation: Minor Revisions Summary

"This manuscript focus on the prolonged and intense Drought episode that struck Chile between 2010 and 2015. The authors provide a comprehensive characterization of this so called Mega-Drought event from various perspectives (meteorological, hydrological, anomalous climate dynamics, vegetation dynamics) but also considering the long-term context (last millennium), and finally providing some framing within regional warming

background. The work is very interesting to read, with plenty of informative figures. The major problems I see are related with the level of novelty of this manuscript taking into account the contents of two other papers by the authors with some overlap in contents. I'm also particularly concerned with the amateurish attitude of the authors in relation to citations with so many missing and wrong references. Thus, I believe the paper can be accepted if the authors improve the manuscript taking into consideration the following clarifications listed below"

R. Thanks for your overall assessment of our work. We plan to fully address your major and specific comments in order improve the quality and readability of our manuscript. Major issues

"1. Level of originality Despite the overall good quality of the work presented here I must confess that an interested reader cannot be entirely sure on the level of originality of the contents included in the manuscript, particularly taking into account the two sister publications carry out by these authors, and covering (at least in part) the same Mega-drought event (Boisier et al., 2016 and Garreaud et al., submitted). While I can understand perfectly well that such a major event can be characterized from multiple perspectives, it is not entirely clear the level of superposition (if any) among these three manuscripts. Please provide a clarification on this important issue."

R. Thanks for pointing this out. The paper by Boisier et al. (2016), published in GRL, focuses on the rainfall trends in this region from the late 70's onward and attributes these trends to both natural variability (ENSO+PDO) and anthropogenic forcing. Of course, the current mega-drought contributed to the existence of this trend, and an attribution exercise was done, but the actual mechanism behind the protracted dry condition was not unveiled. We plan to investigate those mechanism in a paper cited in the HESS manuscript as Garreaud et al. submitted. It turns out that such paper was rejected and we are currently working in a new version including some long numerical simulations. In any case, the soon-to-be manuscript (intended for J. of Climate) only addressed the large-scale atmospheric forcing of the central Chile mega drought (here
is the problem...we know that a major part of the drought was ocean-forced, but we don't know yet which part of the ocean was most influential...at least not the tropical Pacific). In the new version of our HESS manuscript we also plan to cite a handful of published papers on previous droughts in central Chile. None of those past events persisted for more than 3-years. This is precisely a novel aspect of the current event... its very long duration (7 years now).

In sum, the new version will explicitly state that our manuscript is the only scientific work addressing the physical, regional aspects of the central Chile mega drought, an event that is unprecedented in itself.

"2. References The authors were extremely careless with the references. It is unacceptable that you have so many missing and wrong references, including papers by the authors (?). This is quite distracting when a reader is trying to put the scientific questions in context of previous literature."

R. We deeply apologize for this mistake (my mistake!). We pasted an old version of the reference section in the submitted version...this explains the numerous omissions. Of course, the revised version will include a full, updated list of references. We also will take the opportunity to correct several typographical, spelling and usage errors.

"Minor suggestions 1. (Introduction, 2nd parag) The authors introduce several major droughts that have occurred around the world in the last two decades. Taking into account the amount of people affected and the outstanding implications I would like to suggest to add the Middle East (or Fertile Crescent) drought between 2007-2012 (Trigo et al., 2010; Kelly et al., 2005)"

R. References to these analogs will be added. Thanks for pointing them out.

"2. (Page 5, lines 22-23) This statement that winter precipitation corresponds to >75 needs a reference to support it. More importantly, this is not consistent with Fig. 1b where winter precipitation for regions south of 33°S represents less than 75%. Please

rewrite sentence, adapting to the large N-S gradient of winter precipitation contribution"

R. Good point, the 75% winter accumulation is only a reference (this is why we include Fig. 1b) and text will be rephrased.

"3 (Page 6, lines 32-33) The marked West-East gradient in precipitation is not so clear at all latitudes as it happens mostly in the central section of Chile. North of 33°S and South of 38°S it appears to be negligible."

R. The map in Fig. 1b has not enough data to portray the zonal gradient (and actually, there is no much data in the high Andes). Text clarified.

"4 (Page 7, lines 23) I believe this low-order correlation refers to the autocorrelation coefficient? If so please clarify it."

R. Yes, autocorrelation. Corrected.

"5 (Page 8, lines 2-3) If you have continuous precipitation data from 1960 until 2015 why restricting the historical comparison period to the 1961-2000 (40 years) instead of considering 1960-2009 (50 years)?"

R. The auto calibration model to obtain the SPI only needs 40 years, but we have expanded the reference period and results don't change significantly. Commented in the new version.

"6 (Page 8, lines 3,5,9) Although there are no standard procedures, the SPI acronym is usually employed as such, the temporal scales should be added as indices (or brackets). Please consider adapting the cumbersome SPI-12D to SPI12D or even SPI12"

R. Done, we will use SPI12D

"7 (Page 11, lines 27-28) Can you provide some additional information or literature for the Andes regarding the separation of the role played by diminishing precipitation and increasing temperatures in terms of reduced snow pack."

R. We wish we could, but to our knowledge such comparison hasn't been done for the Andes. It will be commented.

"8 (Page 12, line 11) Please check if Fig.8 should be Fig.9 here." R. It actually refer to Fig. 8d. Changed.

"9 (Page 13, line 5) Please check if Fig.9b should be Fig.10b here." R. Thanks for pointing this out...we changed to Fig. 10b

"10 (Page 13, lines 18-22) This increment of DTR is consistent with the remaining of South America? Until the AR4 IPCC report in 2007 the DTR was diminishing in most areas of the world, but that has changed in the last decade. Can you provide a little bit more of regional (remaining S. America) and temporal (changes in DTR trend signal) context."

R. This is an interesting point. Along Chile are the warming trend is very marked in the maximum (daytime) temperatures but much smaller in the minimum (nighttime) temperatures. As you pointed out, this is contrary to most of continental land masses. Right now we only can note and hypothesize on this: during daytime there is entrainment of the free tropospheric air into the mixed layer that otherwise is not warming due to the ocean influence. This will be commented in the new version.

"11 (Page 13, line 25) Please check if Fig.10b should be Fig.11b here." R. Thanks for pointing this out...we changed to Fig. 11b

"12 (Page 13) Please check carefully all Figure numbers. Some appear to be lagging by one (e.g. Figure10c means Figure11c, Figure12b means Figure13b)." Yes, there was a problem with the figure numbers. Problem fixed.

"Figures Fig.1d The scale used is a bit misleading. Every station appears with the same reddish color and it is very difficult to distinguish regions. What is the point of presenting the range of possible values between -1 and 1? Please compress the possible values to the range [0 1] or even [0.5 1]. That will provide a much more informative plot."

R. We modified the color scale (distributed between +0.5 and +1 in new version).

"Fig.5a and Fig.5b. Can you explain how come a few stations present a positive rainfall trend (Fig.5a) or streamflow trend (Fig.5b) in the midst of strong negative trends everywhere else?"

R. We only can speculate in problems with the data. We are doing an extra quality control for them.

Fig.9 Please provide a clear link between each subpanel letter (a,b,c,d,e) and the corresponding section in the figure caption. R. Caption corrected.

"Fig.10. Please provide the meaning of the regression dashed lines in the figure caption." R. Caption corrected.

"Fig.12. If you describe first Fig.12b and then Fig.12a why not provide the information in the logical reverse order, i.e. place the map on the left (becoming Fig.12a) and the scatter plot on the right (becoming Fig.12b)?"

R. Order altered following your advice.

―――――――――――――――――――

---

## Author Comment (AC2) · 15 Jul 2017

Original comments in quotes "...."

Reviewer 1 (Dr. Enenkel) Recommendation: Major Revisions Summary

"The study of Garreaud et al. puts the recent perennial drought conditions in Chile into a multi-dimensional historical and thematic context. The strengths of the manuscript lie in the holistic approach, the level of analytical detail, the visualization of drought conditions on a regional/global scale and the link to long-term climate conditions via tree-ring analysis"

R. Thanks for your overall assessment of our work. We are particularly pleased to see that you recognize the integrative, holistic approach of our work on the Mega Drought. Indeed, we consider this manuscript as an initial work upon which we and other colleagues can build more specific analyses, including the impacts on socio-economic sectors. We are fully addressing your major and specific comments in order improve the quality and readability of our manuscript.

"I agree with my fellow reviewer that, in the context of existing literature about the Chilean MD (even from the author's themselves), the novelty of the manuscript is questionable"

R. Thanks for pointing this out. The paper by Boisier et al. (2016), published in GRL, focuses on the rainfall trends in this region from the late 70's onward and attributes these trends to both natural variability (ENSO+PDO) and anthropogenic forcing. Of course, the current mega-drought contributed to the existence of this trend, and an attribution exercise was done, but the actual mechanism behind the protracted dry condition was not unveiled. We plan to investigate those mechanism in a paper cited in the HESS manuscript as Garreaud et al. submitted. It turns out that such paper was rejected and we are currently working in a new version including some long numerical simulations. In any case, the soon-to-be manuscript (intended for J. of Climate) only addressed the large-scale atmospheric forcing of the central Chile mega drought. In the new version of our HESS manuscript we also plan to cite a handful of published papers on previous droughts in central Chile. None of those past events persisted for more than 3-years. In sum, the new version will explicitly state that our manuscript is the only scientific work addressing the physical, regional aspects of the central Chile mega drought, an event that is unprecedented in itself.

"However, what bothers me more is that the paper implies improved drought preparedness via an "understanding of the nature and biophysical impacts of the MD". Unfortunately, it fails to relate the presented fidings to any kind of decision-making or socio-economic response. I see potential in the paper if the authors manage to link

the tracking of the MD through the season(s) via different variables and the MD's historical/climatic context to any kind of suggestion for decision-support, such as socio-economic countermeasures (e.g. changes in agricultural practices, consideration of seasonal climate forecasts)".

R. We began (and concluded) our manuscript with the statement "Understanding the nature and impacts of the current multiyear drought will thus contribute to our preparedness efforts to face the projected dry, warm regional climate scenarios". Broadly speaking this is true, but we acknowledge that the material presented in this work -by itself- doesn't make a tangible contribution to preparedness efforts. On this basis, we have reworded such sentences to avoid creating too high expectations in this issue. As you mention, linking the "physical" MD to the socio-economic world is a major task and probably deserve other paper. But we are taking the opportunity and the new version provide some elements on the socio-economic response to the protracted dry spell.

First, we now include a table (Fig. 1 here) showing the evolution of the water shortage decrees during the MD period for the administrative regions that conform central Chile. We did a little discussion on this in the new introduction. These decrees are presented by the National Government and allows to alter the normal water rights system. We note a mismatch between the drought intensity (as per the rainfall deficit) and the decrees -both in space and time. For instance, 2013 was the driest year during the MD and has the lower number of decrees.

We also will add information about the expenditures by the central government in trucks to deliver potable water to rural communities. This information is available on a regional basis for the whole period and is summarized in the attached figure 2 (to be included in the new version). The size of each truck is proportional to the 2010-2015 expenditures for each region normalized by population. Here we see a large economic burden in the regions to the north of Santiago (semi arid sector) and around Concepción (where the water deficit was actually largest). Therefore, in contrast to the water emergency decrees, the actual expenditures to alleviate the MD have a better match with the actual

rainfall deficit. These brief discussions will be added to the final part of our paper.

Finally, we include the reference to a new article (Unpacking resilience for adaptation: incorporating practitioners experiences through a transdisciplinary approach, the case of drought in Chile; by Aldunce et al. 2016 in Sustainability), where the author disuses the usefulness of existing drought measurements in Chile to distill strength and weakness of adaptation measurements at the local level. Interesting, the worst evaluated measure are the water shortage decree.

Additionally to the previous points, we plan to conclude the new version of the manuscript with a list of pending issues. This include a more comprehensive quantification of the social and economical impacts of the MD, that will serve as a basis for suggesting decision-support measures. Even without such quantification, a real time monitoring of drought state appears as a key element, set aside the prospect of inter-annual prediction of hydrological conditions. Another big issue is the quantification of the use (overuse?) of ground water. In the paper we note a lack of "browning" and even "greening" of the vegetation over parts of central Chile during the MD and that may be the result of intense use of groundwater. The key question is whether or not that resources are being used in a sustainable manner.

"I understand that an in-depth consideration of the manuscript's fidings related to decision-support or climate-change adaptation is out of scope, but even a superficial discussion would improve the manuscript's"

R. We thanks again for your suggestion. As you can see we plan to add four elements to our discussion (water shortage decrees, expenditures on water distributed by truck, reference to a new paper documenting usefulness of existing drought measurements) and a listing of pending issues. Granted, these new elements in our paper only address partially the social impacts of the MD, but we hope they will a good addition to this manuscript in HESS.

Main comments "The manuscript needs to be more clearly distinguished from the authors' other publications" R. Yes...please see our response to the second point of your overall evaluation

"References and figures need to be reviewed (some are missing, some only mentioned in the reference list, but not in the text)"

R. We deeply apologize for the mistake with the reference section. Of course, the revised version will include a full, updated list of references. We also will take the opportunity to correct several typographical, spelling and usage errors.

"Although 2010 was the strongest La Niña year during the MD period 2011, 2013 and 2014 were La Niña years as well. Could these conditions have contributed to the MD's persistence?"

R. Considering the Niño3.4 index, the winter of 2011, 2013 and 2014 were rather neutral. In any case, the relatively cold condition of the Pacific in this period (cold PDO phase) was probably an important forcing of the MD period. Part of this is addressed in Boisier et al. 2016.

General comments "Whenever you talk about rainfall deficits and related percentages please mention the reference period." R. Advice taken. Reference period added.

"Instead of describing selected drought events in the introduction I suggest you provide more general statistics, if available (e.g. from UNISDR; EMDAT might not be a good choice)"

R. Not sure which drought events. We think that mention a few examples of recent past droughts in other subtropical regions is necessary since it gives a background for our event.
* * *
[Figure]

| | 2008 | 2009 | 2010 | 2011 | 2012 | 2013 | 2014 | MS |
|---|---|---|---|---|---|---|---|---|
| Coquimbo | 3 | 0 | 2 | 3 | 1 | 2 | 4 | 15 |
| Valparaiso | 3 | 0 | 2 | 5 | 4 | 3 | 5 | 22 |
| Metropolina | 2 | 0 | 0 | 2 | 2 | 0 | 0 | 6 |
| O'Higgins | 1 | 0 | 0 | 1 | 0 | 0 | 0 | 2 |
| Maule | 1 | 0 | 0 | 2 | 1 | 1 | 2 | 7 |
| Bio Bio | 1 | 1 | 0 | 0 | 0 | 0 | 1 | 3 |
| Total | 11 | 1 | 4 | 13 | 8 | 6 | 12 | 55 |

**Fig. 1.**

**Fig. 2.**

---

## Author Response (AR1)

Santiago, September 7, 2017

Dr. Micha Werner
Associated Editor
*Hydrology and Earth System Sciences* (*HESS*)

Dear Dr. Werner:

We have finished the revision of our manuscript "The 2010-2015 mega drought in Central Chile: Impacts on regional hydroclimate and vegetation" submitted for publication in *HESS*.

We are grateful of the constructive criticism that Dr. Trigo (reviewer 1) and Dr. Enenkel (reviewer 2) formulated on our work. In our opinion, both reviewers have a favorable view of our manuscript. Yet, they expressed general/major comments that helped us to generate a scientifically stronger version. The reviewer's comments also helped us to add key references, as well as correct several typographical, spelling and usage errors.

The following pages contain a detailed point-by-point reply to the questions and comments (comments in brown, answers in black). Below is a summary of the main changes:

> We expressed more clearly the originality of our work. Indeed, to date, there is only one published paper that address, only partially, the large-scale atmospheric context of the central Chile mega drought (Boisier et al. 2016) and two technical reports written by our group (CR2 2015) and the Ministry of Public Work.

- We have added a discussion (new section 7, Figs. 14 and S6) on the link between the "physical" drought (i.e., actual changes in hydrological variables) and its impacts on society and responses from the State. A full description of those effects is beyond the scope of the present work and not suitable for HESS, but, as suggested by Dr. Enenkel, a cursory inspection of these aspects made the manuscript more appealing to broader community.

- We did a proof reading of the manuscript. We apologize for the problem with the reference section (many references were missing), a mistake from my part. We also improved some figures to enhance its clarity.

We are confident that the new version will be more interesting and accessible to the readers of *HESS*.

We look forward to hearing from you,

**René Garreaud**
Universidad de Chile

**Reviewer 1 (Dr. Trigo)**

Recommendation: Minor Revisions

**Summary**

This manuscript focus on the prolonged and intense Drought episode that struck Chile between 2010 and 2015. The authors provide a comprehensive characterization of this so called Mega-Drought event from various perspectives (meteorological, hydrological, anomalous climate dynamics, vegetation dynamics) but also considering the long-term context (last millennium), and finally providing some framing within regional warming background. The work is very interesting to read, with plenty of informative figures. The major problems I see are related with the level of novelty of this manuscript taking into account the contents of two other papers by the authors with some overlap in contents. I'm also particularly concerned with the amateurish attitude of the authors in relation to citations with so many missing and wrong references. Thus, I believe the paper can be accepted if the authors improve the manuscript taking into consideration the following clarifications listed below.

Thanks for your overall assessment of our work. In the revised version we addressed your major and specific comments in order improve the quality and readability of our manuscript.

Major issues

1. Level of originality

Despite the overall good quality of the work presented here I must confess that an interested reader cannot be entirely sure on the level of originality of the contents included in the manuscript, particularly taking into account the two sister publications carry out by these authors, and covering (at least in part) the same Mega-drought event (Boisier et al., 2016 and Garreaud et al., submitted). While I can understand perfectly well that such a major event can be characterized from multiple perspectives, it is not entirely clear the level of superposition (if any) among these three manuscripts. Please provide a clarification on this important issue.

Thanks for pointing this out. The paper by Boisier et al. (2016), published in *GRL*, focuses on the rainfall trends in this region from the late 70's onward and attributes these trends to both natural variability (ENSO+PDO) and anthropogenic forcing. Of course, the current mega-drought contributed to the existence of this trend, and an attribution exercise was done, but the actual mechanism behind the protracted dry condition was not unveiled. We plan to investigate those mechanism in a paper cited in the HESS manuscript as *Garreaud et al. submitted*. It turns out that such paper was rejected and we are currently working in a new version including some long numerical simulations.

In any case, Boisier et al. (2016) and the soon-to-be manuscript (intended for *J. of Climate*) only address the large-scale atmospheric forcing of the central Chile mega drought. By the contrary, the present work in HESS describe the mega-drought, place it in historical and longer-term context and document some of their impacts. No such integrative effort has been published but for a technical (non-scientific) report from our own group (the CR2 report to the nation, 2016).

The novelty of the present work is emphasized in the revised version. Please see lines 28-32 (page 3), lines 4-10 (page 4) and lines 25-29 (page 11).

Yes, there was a problem with the figure numbers. The problem was fixed in the new version.

Figures

Fig.1d The scale used is a bit misleading. Every station appears with the same reddish color and it is very difficult to distinguish regions. What is the point of presenting the range of possible values between -1 and 1? Please compress the possible values to the range [0 1] or even [0.5 1]. That will provide a much more informative plot.

We modified the color scale (distributed between +0.5 and +1 in new version) in new Figure 1(d).

Fig.5a and Fig.5b. Can you explain how come a few stations present a positive rainfall trend (Fig.5a) or streamflow trend (Fig.5b) in the midst of strong negative trends everywhere else?

We only can speculate in problems with the data. It is commented in the caption of the new Figure 5.

Fig.9 Please provide a clear link between each subpanel letter (a,b,c,d,e) and the corresponding section in the figure caption.

Caption corrected.

Fig.10. Please provide the meaning of the regression dashed lines in the figure caption.

Caption corrected.

Fig.12. If you describe first Fig.12b and then Fig.12a why not provide the information in the logical reverse order, i.e. place the map on the left (becoming Fig.12a) and the scatter plot on the right (becoming Fig.12b)?

Order of the panels altered following your advice.

**Reviewer 2 (Dr. Enenkel)**

Recommendation: Major Revisions

Summary

The study of Garreaud et al. puts the recent perennial drought conditions in Chile into a multi-dimensional historical and thematic context. The strengths of the manuscript lie in the holistic approach, the level of analytical detail, the visualization of drought conditions on a regional/global scale and the link to long-term climate conditions via tree-ring analysis.

Thanks for your overall assessment of our work. We are particularly pleased to see that you recognize the integrative, holistic approach of our work on the Mega Drought. Indeed, we consider this manuscript as an initial work upon which we and other colleagues can build more specific analyses, including the impacts on socio-economic sectors.

We have addressed your major and specific comments in order improve the quality and readability of our manuscript.

I agree with my fellow reviewer that, in the context of existing literature about the Chilean MD (even from the author's themselves), the novelty of the manuscript is questionable.

Thanks for pointing this out. The paper by Boisier et al. (2016), published in *GRL*, focuses on the rainfall trends in this region from the late 70's onward and attributes these trends to both natural variability (ENSO+PDO) and anthropogenic forcing. Of course, the current mega-drought contributed to the existence of this trend, and an attribution exercise was done, but the actual mechanism behind the protracted dry condition was not unveiled. We plan to investigate those mechanism in a paper cited in the HESS manuscript as *Garreaud et al. submitted*. It turns out that such paper was rejected and we are currently working in a new version including some long numerical simulations.

In any case, Boisier et al. (2016) and the soon-to-be manuscript (intended for *J. of Climate*) only address the large-scale atmospheric forcing of the central Chile mega drought. By the contrary, the present work in HESS describe the mega-drought, place it in historical and longer-term context and document some of their impacts. No such integrative effort has been published but for a technical (non-scientific) report from our own group (the CR2 report to the nation, 2016).

The novelty of the present work is emphasized in the revised version. Please see lines 28-32 (page 3), lines 4-10 (page 4) and lines 25-29 (page 11).

However, what bothers me more is that the paper implies improved drought preparedness via an "understanding of the nature and biophysical impacts of the MD". Unfortunately, it fails to relate the presented findings to any kind of decision-making or socio-economic response. I see potential in the paper if the authors manage to link the tracking of the MD through the season(s) via different variables and the MD's historical/climatic context to any kind of suggestion for decision-support, such as socio-economic countermeasures (e.g. changes in agricultural practices, consideration of seasonal climate forecasts).

I understand that an in-depth consideration of the manuscript's findings related to decision-support or climate-change adaptation is out of scope, but even a superficial discussion would improve the manuscript's

Your comment was quite challenging for our team. We fully agree that linking bio-physical aspects of the MD to socio-economic aspects is utterly important if one pretends to contribute to "preparedness efforts to face the projected dry, warm regional climate scenarios" (and generally, new droughts). Nonetheless, we don't have the data (nor the expertise) to conduct a comprehensive study of this link, neither HESS seems the more suitable journal for that.

After careful consideration, we decided to gather as much social/economic information as possible and describe it in the new Discussion section (pages 15-16-17). We believe that the material in the new section satisfies the need for a "superficial discussion" that you proposed and, in retrospective, we are pleased to do so because our paper becomes more attractive to a broader audience. Rather than having final answers, our discussion section propose a number of outstanding questions (e.g., assessment of the overall economic impacts, evolution of groundwater).

The discussion section describe the impacts of the mega drought on society by considering a few recent papers on public perception and technical reports by sanitary and energy companies on their services and cost during the MD (lines 1-10, page 16; Figure S6). As reported elsewhere, rural communities faced a far worse scenario than cities, with shortage of water for drinking and agriculture. We then review the responses by the state that involve (at least) three types of measures: intervention of the water market, agriculture emergency and delivering potable water by truck to rural communities (lines 9-29, page 16). We were able to obtain information on these measures disaggregated by year and administrative region, as summarized in new Fig. 14. Interesting, there is an evident mismatch between these state initiatives and the actual rainfall deficit (but for the water-by-truck expenditures), suggesting that criteria other than the actual lack of water is being used by the policy makers. We also describe the drought monitoring systems implemented in Chile (lines 30-37, page 16).

Please read the new section 7 (Discussion) and take a look at new Figs. 14 and S6. Finally, the original manuscript began (and concluded) with the statement "Understanding the nature and impacts of the current multiyear drought will thus contribute to our preparedness efforts to face the projected dry, warm regional climate scenarios". Broadly speaking this is true, but we acknowledge that the material presented in this work -by itself- doesn't make a tangible contribution to preparedness efforts. On this basis, we have reworded such sentences to avoid creating too high expectations in this issue (lines 15-16, page 2; lines 7-14 page 18).

**Main comments**

**The manuscript needs to be more clearly distinguished from the authors' other publications**

Yes...please see our response to the second point of your overall evaluation

**References and figures need to be reviewed (some are missing, some only mentioned in the reference list, but not in the text)**

We deeply apologize for the mistake with the reference section. The revised version include a full, updated list of references. We also will take the opportunity to correct several typographical, spelling and usage errors.

**Although 2010 was the strongest La Niña year during the MD period 2011, 2013 and 2014 were La Niña years as well. Could these conditions have contributed to the MD's persistence?**

Considering the Niño3.4 index, the winter of 2011, 2013 and 2014 were rather neutral. In any case, the relatively cold condition of the Pacific in this period (cold PDO phase) was probably an important forcing of the MD period. Part of this is addressed in Boisier et al. 2016 (see lines 18-20, pages 11).

**General comments**

**Whenever you talk about rainfall deficits and related percentages please mention the reference period.**

Advice taken. The reference period (1970-2010) for climate analysis is now stated in the data description section (see lines 4-5, page 5). Other reference periods are noted in the figure captions.

Instead of describing selected drought events in the introduction I suggest you provide more general statistics, if available (e.g. from UNISDR; EMDAT might not be a good choice)

Mentioning a few examples of recent past droughts in other subtropical regions is necessary as a background for our event. Reviewer 1 gave a few more examples that were included in the new version (lines 10-19, page 3).

**The following pages are the marked-up revised manuscript**

[revised manuscript text omitted]

---

## Author Response (AR2)

Santiago, November 7, 2017

Dr. Micha Werner
Associated Editor
*Hydrology and Earth System Sciences* (*HESS*)

Dear Dr. Werner:

We have finished the final version of our manuscript "The 2010-2015 mega drought in Central Chile: Impacts on regional hydroclimate and vegetation" submitted for publication in *HESS*.

We are pleased that you found that our revised version (September 2017) properly addressed the comments raised by Dr. Trigo (reviewer 1) and Dr. Enenkel (reviewer 2). In retrospective, those comments helped us to produce a more comprehensive manuscript, by including few but key aspects of the drought's human dimension (perception and responses).

We are also very grateful of your detailed revision of the manuscript. All you comments were addressed in the final version. The following pages contain a detailed point-by-point reply to the questions and comments (comments in brown, answers in black).

We are confident that the new version will be more interesting and accessible to the readers of *HESS*.

Looking forward to hear from you,

**René Garreaud**
Universidad de Chile

**Editor Comments**

Recommendation: Minor Revisions

**Summary**

I have carefully read the revised manuscript. I think the comments raised by the reviewers have been adequately addressed, without substantially changing the scope of the paper (this refers primarily at the socio-economic impacts of the Mega-Drought), and would propose to now move forward to publication. However, on reading the manuscript I encountered several minor editorial issues. I have annotated the revised manuscript, and would like to request the authors to address these issues in a final revision of the manuscript.

Thanks for your overall assessment of our work, and the improvements incorporated in the revised version.

In the final version we addressed **all** your minor editorial issues (see tracked version) in order improve the quality and readability of our manuscript. I don't think that listing all these comments is necessary, but let me respond those "majors" :

Page 7, Comment 3: This sentence is somewhat unclear. How is the median used in determining the RPI? Please clarify

The RPI is defined more clearly. See Page 7, line 29.

Page 8, Comment 1: This is in fact 25.5% of the time - so closer to a quarter - which is somewhat trivial as it is the 25% deficit

That correct, droughts of that magnitude occur in a quarter of the time (Page 8, line 5).

Page 9, Comment 1: why were thresholds of -0.3 and -1.1 chosen? Previously the thresholds was chosen a -1. This new selection seems somewhat arbitrary

Corrected. The actual thresholds are -1.0 (as previously used) and -0.3 (a minimum threshold for dry periods). See page 9, line 7.

Page 12, comment 6: In the figure the caption notes that this is the dynamic level - should it not be the static level? To my mind the dynamic level is the level when pumping is active. Is the Alfalfares well an operational well?

We verified that we are showing the static level. The figure legend was corrected.

Page 13, Commment 4: I think this is somewhat suggestive. While I agree that the concept of stationary hydrology may be challenged - I think the comparison of the prolonged MD event to previous shorter drought periods does not substantiate the claim - for that the a change in response in future shorter drought periods would need to be assessed

We agree with your comment. We will have to wait a few years (decades) to see if there was a permanent change in hydrology. We modified this sentence. See page 13, line 28.

Page 18, comment 1: This does not seem consistent - in previous text it is suggested that shrublands in the Northern area were affected - but impacts to pasture lands in the South less marked - please check.

Corrected. See page 18, lines 2-3.

The following pages are the marked-up revised manuscript

[revised manuscript text omitted]